# Clu1/Clu form mitochondria-associated granules upon metabolic transitions and regulate mitochondrial protein translation via ribosome interactions

Leonor Miller-Fleming*, Wing Hei Au[¤a], Laura Raik, Pedro Rebelo-Guiomar[¤b], Jasper Schmitz, Ha Yoon Cho, Aron Czako, Alexander J. Whitworth*

MRC Mitochondrial Biology Unit, University of Cambridge, Cambridge Biomedical Campus, Cambridge, United Kingdom

☯ These authors contributed equally.
¤a Current address: John van Geest Centre for Brain Repair, Department of Clinical Neurosciences, University of Cambridge, Cambridge Biomedical Campus, Cambridge, United Kingdom
¤b Current address: Department of Biochemistry, University of Cambridge, Tennis Court Road, Cambridge, United Kingdom
* leonor.miller.fleming@gmail.com (LM-F); ajw69@cam.ac.uk (AJW)

## Abstract

Mitochondria perform essential metabolic functions and respond rapidly to changes in metabolic and stress conditions. As the majority of mitochondrial proteins are nuclear-encoded, intricate post-transcriptional regulation is crucial to enable mitochondria to adapt to changing cellular demands. The eukaryotic Clustered mitochondria protein family has emerged as an important regulator of mitochondrial function during metabolic shifts. Here, we show that the *Drosophila melanogaster* and *Saccharomyces cerevisiae* Clu/Clu1 proteins form dynamic, membraneless, mRNA-containing granules adjacent to mitochondria in response to metabolic changes. Yeast Clu1 regulates the translation of a subset of nuclear-encoded mitochondrial proteins by interacting with their mRNAs while these are engaged in translation. We further show that Clu1 regulates translation by interacting with polysomes, independently of whether it is in a diffuse or granular state. Our results demonstrate remarkable functional conservation with other members of the Clustered mitochondria protein family and suggest that Clu/Clu1 granules isolate and concentrate ribosomes engaged in translating their mRNA targets, thus, integrating metabolic signals with the regulation of mitochondrial protein synthesis.

## Author summary

Mitochondria are essential cellular organelles that perform many important roles in regulating metabolism. They are dynamic in form, function and composition

**Data availability statement:** All relevant data are within the manuscript and its Supporting Information files.

**Funding:** This work was supported by European Research Council Starting Grant (309742 to AJW), Medical Research Council (MRC) core funding (MC_UU_0028/6 to AJW), MRC project grants (MR/V003933/1 to LMF and AJW), and Motor Neurone Disease Association grant (Whitworth/Apr17/857-79 to LMF and AJW). The funders had no role in study design, data collection and analysis, decision to publish, or preparation of the manuscript.

**Competing interests:** The authors have declared that no competing interests exist.

which is crucial to maintain highly energy-demanding tissues. However, the factors that coordinate the proteomic changes and how they are regulated remain unclear. Here, we show that the orthologous RNA-binding proteins Clu1 (in yeast) and Clu (in *Drosophila*) form dynamic, membraneless, mRNA-containing granules adjacent to mitochondria under changing metabolic conditions. Clu1 regulates the translation of a subset of nuclear-encoded mitochondrial proteins by interacting with their mRNAs while these are engaged in translation. We further show that Clu1 regulates translation by interacting with polysomes. Interestingly, this appears to be independent of whether it is in a diffuse or granular state. Our results demonstrate functional conservation with other members of the Clustered mitochondria protein family and suggest that Clu/Clu1 granules isolate and concentrate ribosomes engaged in translating their mRNA targets, thus, integrating metabolic signals with the regulation of mitochondrial protein synthesis.

## Introduction

Mitochondria perform many essential cellular functions, most notably in the production of ATP and regulation of cell death [1,2]. To perform these myriad functions, mitochondria are extremely dynamic organelles, trafficking along the cytoskeletal tracks, undergoing fission and fusion events, and interacting with other organelles [3]. This dynamic nature allows mitochondria to respond rapidly to changes in cellular conditions, such as transient stresses or fluctuations in nutrient availability [4]. Consequently, disruption of mitochondrial function causes wide-ranging cellular defects and human diseases [5].

While mitochondria contain their own genome and translation machinery, the majority of their proteome is nuclear-encoded, transported to mitochondria, imported and sorted to their appropriate destination [6]. The central role of various mitochondrial activities in multiple cellular and metabolic processes means their protein composition is also highly dynamic [7,8]. Thus, transcriptional and translational changes must be tightly regulated and coordinated between nuclear and mitochondrial genomes, particularly in response to metabolic demands. An essential layer of regulation occurs at the post-transcriptional level, where RNA-binding proteins play a crucial role [9,10]. These proteins are essential for the stability, localisation, and translation of mRNAs, and their dysregulation can underlie different diseases [10].

The Clustered mitochondria protein family is a conserved group of proteins found among eukaryotes, which includes the yeast Clustered mitochondria (Clu1) [11], the *Drosophila* Clueless (Clu) [12] and the mammalian Clustered mitochondria homologue (CLUH) [13] proteins, which share a common mitochondrial clustering phenotype and mitochondrial dysfunction when mutated [11–14]. Knockout mice for *Cluh* are neonatal lethal, while fly mutants for *Clu* mostly die during development, with rare adult flies surviving only a few days [12,15,16], highlighting the critical roles these proteins play in different organisms during crucial developmental and metabolic transitions.

Mammalian CLUH is an RNA-binding protein that preferentially binds and regulates mRNAs of multiple nuclear-encoded mitochondrial proteins involved in the tricarboxylic acid cycle (TCA) cycle, oxidative phosphorylation (OXPHOS) and several other mitochondrial metabolic pathways [13]. However, how this regulation is achieved is still unclear. Emerging evidence has shown that these proteins form foci under different metabolic conditions suggesting these structures are membraneless organelles which may play an important role into how the Clustered mitochondria proteins function [17,18].

Here, we exploited the tractability of *Drosophila* and budding yeast as model systems and investigated in detail the cellular and molecular dynamics of Clu/Clu1 granules, their nature and composition, analysed the yeast Clu1 binding partners *in vivo*, and explored the relationship of Clu1 with mRNA translation. We propose a model where Clu/Clu1 granules isolate ribosomes engaged in translating their target mRNAs in response to metabolic changes, regulating their translation.

## Results

### Clu/Clu1 forms dynamic foci upon metabolic transitions in flies and yeast

Clu was recently reported to have a dynamic localisation in *Drosophila* egg chambers, affected by nutritional status [18]. We independently observed that a line expressing GFP-tagged endogenous Clu (GFP-Clu) exhibited a punctate subcellular distribution in various fly tissues (S1A Fig). Notably, we observed that Clu localisation was especially dynamic in *Drosophila* egg chambers (Fig 1A). After fasting overnight (16 h), GFP-Clu was diffuse in the cytoplasm of nurse cells and follicle cells (Fig 1A). When flies were refed, GFP-Clu rapidly formed distinct foci within 30 min, which gradually increased in size over time (Fig 1A). After 6 h of refeeding, these puncta reached remarkable sizes, particularly in nurse cells, with an average area of 2.37 μm² (up to 40 μm²) (Figs 1A and S1B-E). Interestingly, when flies were 're-fasted', Clu rapidly returned to the diffuse state within minutes (Fig 1A). These results show that Clu localisation is dynamic and regulated by metabolism.

*Saccharomyces cerevisiae* is very well-defined metabolically, and the metabolic state can easily be manipulated by changing nutrient availability in the media. Thus, we assessed whether the yeast Clu orthologue, Clu1, which to date is poorly characterised, also showed a dynamic localisation. For this, we used a GFP-tagged endogenous *CLU1* strain (Clu1-GFP) grown under different metabolic conditions. In the logarithmic (log) phase, characterised by exponential growth and fermentative metabolism, Clu1-GFP was mainly diffuse in the cytoplasm with some cells showing some undefined small puncta (Figs 1B and S1F), as previously reported [18]. Upon transition to the post-diauxic phase (PD), when glucose is exhausted and cells undergo a metabolic transition into respiration (S1F Fig), Clu1-GFP progressively formed very bright, distinct foci, such that nearly all cells (98%) had Clu1-GFP foci 10 h after the diauxic shift (Figs 1B and S1F). This suggested that Clu1 forms foci when metabolism becomes reliant on mitochondrial respiration.

To confirm whether the transition from fermentation to respiration triggered Clu1 foci formation, we switched cells grown in glucose media to media containing respiratory carbon sources such as ethanol, glycerol and galactose. These abrupt metabolic transitions also led to the formation of Clu1-GFP foci (Fig 1C), supporting that the transition to respiration is the trigger for the formation of foci.

To understand whether this phenomenon was reversible, similar to what we observed in *Drosophila* egg chambers, we added back glucose to PD cells. Within minutes, Clu1-GFP foci dissipated and the protein became diffuse throughout the cytoplasm (Fig 1D and S1 Video). In addition, we also found that Clu1 foci dissipate upon removing the carbon source (ethanol) from the media of PD cultures (Fig 1E), suggesting that foci require ATP to be maintained. Together, these results demonstrate that Clu1 localisation is extremely dynamic in yeast, forming slowly upon the metabolic transition from fermentation to respiration, and rapidly disassembling upon reverting their metabolism to fermentation.

### Clu1 foci are distinct from PBs and SGs and form upon mitochondrial stress

Stress granules (SGs) and Processing bodies (P bodies; PBs) are cytoplasmic membraneless organelles, also known as biomolecular condensates, composed of RNAs and proteins. SG formation is induced, while PBs become more abundant, under different stresses including nutritional stress [19,20]. As PBs and SGs in yeast form in the PD and stationary phases

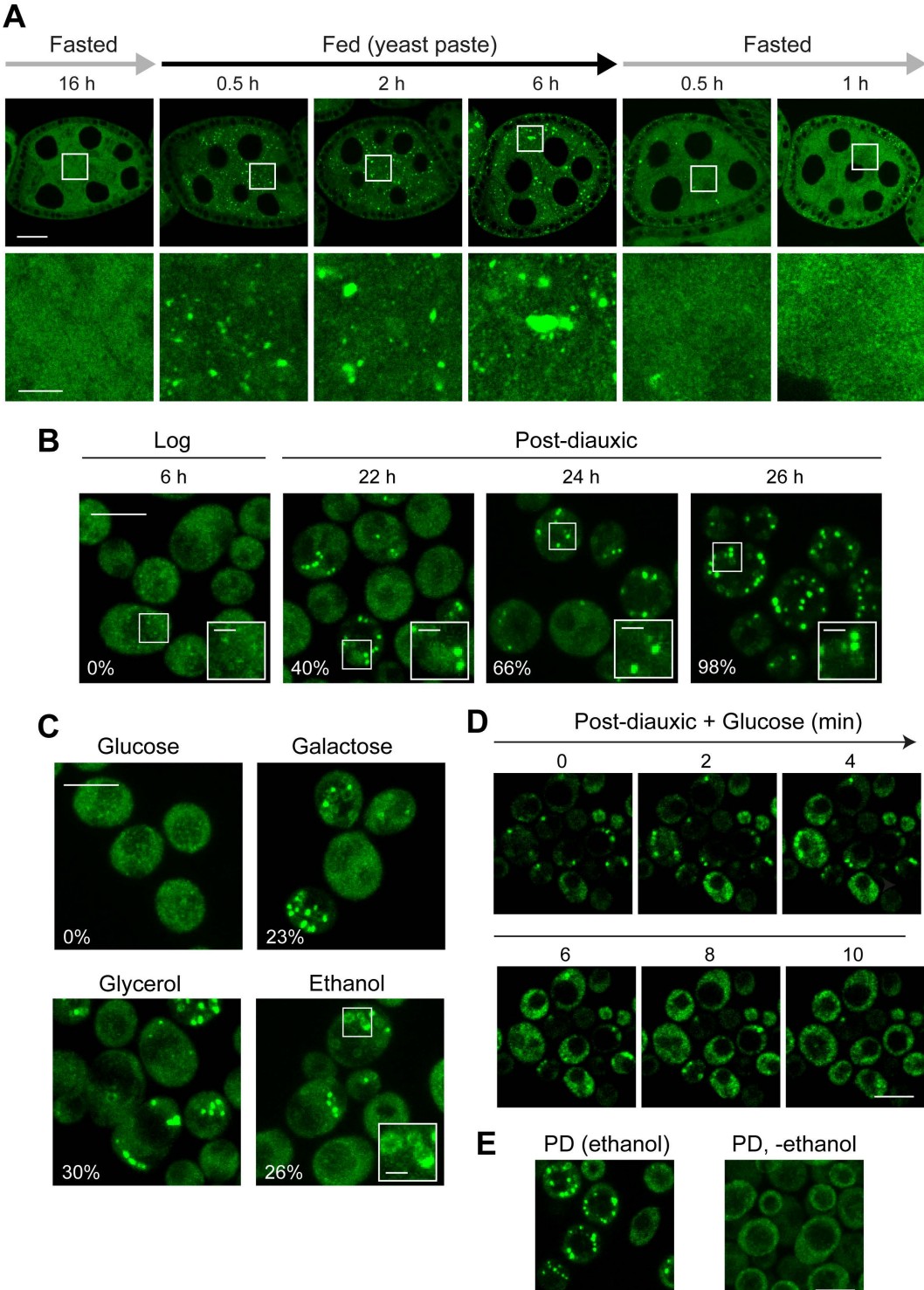

**Fig 1. *Drosophila* Clu and yeast Clu1 form dynamic foci.** (A) Confocal imaging of egg chambers from GFP-Clu flies. One-day-old female flies were mated and fed with yeast paste for 2 days, fasted overnight (16 h, only H2O), refed for 6 h, and re-fasted (only H2O) for 1 h. Images represent the indicated time points during the time-course. (B) BY4741 yeast cells expressing endogenously tagged Clu1-GFP were grown in media containing glucose and analysed by confocal imaging in exponential (log) and post-diauxic (PD) phases. White box regions show Clu1 foci magnified (inset). (C)

Mid-log phase cells grown in media containing glucose were washed and incubated with media containing galactose, glycerol or ethanol and analysed by confocal microscopy after 6 h. White box region (inset) shows Clu1-GFP in a ring-like pattern magnified. (B, C) Numbers indicate the percentage of cells showing large foci from a representative experiment (>200 cells counted). (D) PD Clu1-GFP cells were layered on an agarose pad made with media containing glucose and immediately imaged over a 10 min time course. (E) PD Clu1-GFP cells were washed and incubated in SC media without any carbon source (-ethanol) for 20 min. Cells were imaged and compared with cells kept in the original media (ethanol). (A-E) Representative images of at least three biological replicates. Scale bars: A, top = 20 μm, bottom = 5 μm; B-E, main image = 5 μm; inset = 1 μm.

respectively [21], and are quickly reversed by the addition of glucose, we questioned whether Clu1 was a component of these structures.

First, we tested whether the Clu1 foci appeared upon stresses known to trigger SG and/or PB formation: heat shock, carbon starvation, hypotonic stress, hyperosmotic stress, oxidative stress, and mitochondrial stress (sodium azide) [19–22]. In contrast to SGs and PBs, Clu1 foci were not formed in most tested conditions (Fig 2A). Interestingly, treatment with sodium azide, a complex IV inhibitor and mild uncoupler, rapidly induced the formation of distinct Clu1-GFP foci, resembling the foci observed upon the transition to respiration (Fig 2A). Notably, this induction occurred without any detectable changes in Clu1 protein levels under sodium azide treatment (S2 Fig). Curiously, another mitochondrial stressor, carbonyl cyanide m-chlorophenyl hydrazone (CCCP), a protonophore that dissipates the mitochondrial membrane potential, did not trigger formation of Clu1 foci under this growth condition (Fig 2A).

We also investigated the distribution of Clu1 when cells were more reliant on respiration. For this, we took Clu1-GFP cell cultures that had been shifted from glucose- to galactose-containing media, in which foci started to be formed, and subjected them to heat shock, oxidative stress or mitochondrial stressors. Similar to fermenting conditions, neither heat shock nor oxidative stress further triggered the formation of Clu1 foci. In contrast to what was described for Clu foci in flies [18], Clu1 foci are not sensitive to oxidative stress as this treatment did not lead to their disappearance (Fig 2B). Interestingly, treatments with the mitochondrial stressors sodium azide, CCCP or oligomycin and antimycin A (inhibitors of mitochondrial ATP synthase and complex III, respectively) enhanced the formation of Clu1 foci in all cells (Fig 2B). Spiking glucose into respiring cells grown with galactose and treated with CCCP led to the disappearance of Clu1 foci (Fig 2C), showing the foci are also reversible following mitochondrial stress.

SGs and PBs vary their composition according to the type of stress that leads to their formation [19,20,23]. Thus, although yeast Clu1 foci did not form under most typical stress conditions that trigger SGs and PBs, this did not exclude the possibility that Clu1 could be a component of these structures, specifically during metabolic shift or mitochondrial stress. To test this hypothesis, we co-expressed Clu1-GFP with Ded1-mCherry, a marker of SGs [24], or Dcp1-DsRed, a marker of PBs [25]. We analysed whether these markers co-localise under conditions where both Clu1 foci and SGs or PBs are formed, specifically during the PD phase and after treatment with sodium azide. None of these conditions led to the co-localisation of Clu1 foci with these SG or PB markers (Fig 2D and 2E). It is worth noting that the number of cells with PBs started increasing in the late log phase, as previously described [21], while Clu1 foci only appeared later in the PD phase (Fig 2E). These results support that Clu1 is not a component of SGs or PBs.

Similarly, in fly egg chambers, GFP-Clu foci did not colocalise with SG or PB markers, Fmr1 immunostaining and Tral-mRFP, respectively (S3 Fig), as previously reported [18]. Thus, our results show that despite some similarities between Clu1 foci and SGs and PBs, yeast and *Drosophila* Clu1/Clu foci are different structures.

## Clu foci are biomolecular condensates adjacent to mitochondria

While analysing yeast Clu1 localisation in PD cells, we consistently observed Clu1 foci localised near mitochondria, mostly between adjacent mitochondria (Fig 3B-D). In a few cells, Clu1 appeared in a ring shape surrounding a mitochondrion (Figs S4A and 1C). We also observed a similar mitochondria-adjacent distribution of Clu1 foci when they formed under a different respiratory stress (cells shifted to galactose and treated with CCCP) (Fig 3E), indicating that a different trigger of

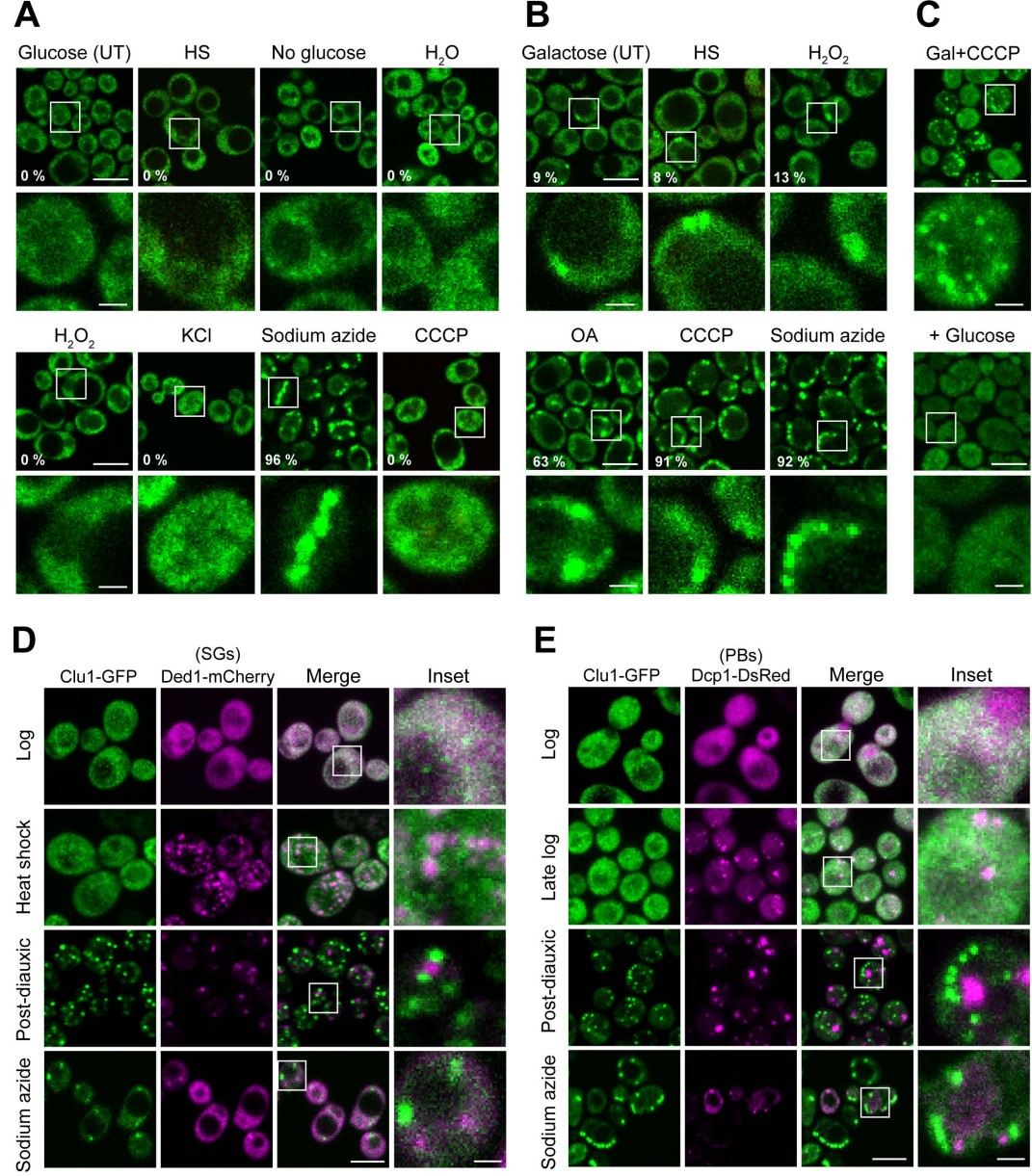

**Fig 2. Yeast Clu1 form foci under mitochondrial stresses and do not co-localise with PBs and SGs.** (A) Maximum intensity projection of confocal images of Clu1-GFP cells grown in glucose-containing media until mid-log phase, then left untreated (UT) or subjected to: heat shock (HS; 46 °C, 30 min), carbon starvation (No glucose, 10 min), hyperosmotic stress (1 M KCl, 30 min), hypotonic stress (H2O, 10 min), oxidative stress (3 mM H2O2, 15 min), and mitochondrial stress sodium azide (0.5% (v/v), 15 min) and CCCP (30 μM, 15 min). (B) Confocal imaging (maximum intensity projection) of mid-log Clu1-GFP cells shifted from glucose to galactose-containing-media and incubated for 6 h, then treated as in A, plus oligomycin (10 μM) and antimycin A (40 μM) (OA; 30 min). (A, B) Numbers indicate the percentage of cells showing large foci from a representative experiment (>200 cells counted). (C) Maximum intensity projection of Clu1-GFP cells grown in galactose-containing media for 6 h, treated with CCCP for 15 min (Gal+CCCP), spiked with glucose and imaged 20 min later (+ Glucose). (D) Confocal microscopy of Clu1-GFP strain expressing Ded1-mCherry, a marker of stress granules (SGs) in the log and PD phases, and after heat shock or sodium azide treatment, as described above. (E) Confocal microscopy images of Clu1-GFP strain expressing Dcp1-DsRed, a marker of P-bodies (PBs) in the log, late log and PD phases, and after sodium azide treatment. Scale bars = 5 μm; insets = 1 μm.

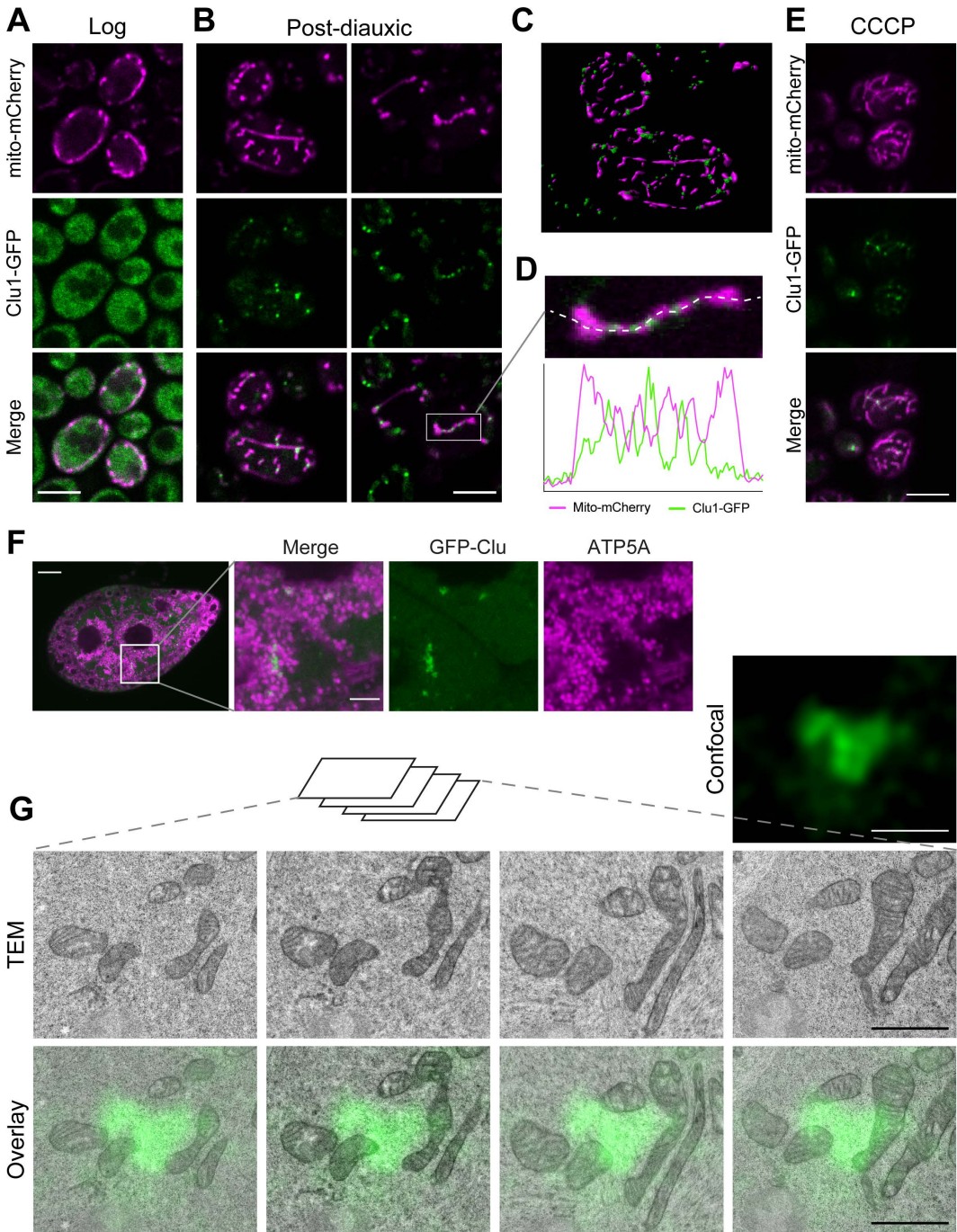

**Fig 3. Clu1/Clu foci are localised in close proximity to mitochondria in yeast and flies.** (A, B) Confocal microscopy of yeast Clu1-GFP cells co-expressing the mitochondrial marker mito-mCherry in the log (A) and PD phases (B). (C) 3D rendering of cells in PD phase. (D) Inset of panel B with an intensity profile plot along the dashed line showing the fluorescence distribution of Clu1-GFP and mito-mCherry. (E) Confocal image of Clu1-GFP cells expressing mito-mCherry after being shifted to galactose for 6 h and treated with CCCP. (F) Confocal imaging of a GFP-Clu egg chamber from three-day-old flies (6 h of feeding after overnight fasting) immunostained for ATP5A to visualise mitochondria. (G) Correlative light and electron microscopy images of a GFP-Clu egg chamber nurse cell showing the juxtaposition of a GFP-Clu foci and mitochondria. The serial transmission electron microscopy (TEM) images, taken at 5000 × magnification, correspond to the confocal image shown. Scale bars: A, B, E = 5 μm; F = 20 μm, inset = 5 μm; G = 1 μm.

foci formation leads to a similar behaviour. The close association of Clu/Clu1 foci with mitochondria and their response to changing metabolic conditions is consistent with Clu/Clu1 performing a role in maintaining mitochondrial function during metabolic changes. Indeed, while knockout of *CLU1* in yeast (*clu1Δ*) had no impact on growth in glucose-containing media (S4B Fig), viability and growth were affected in ethanol-containing media, when respiration is essential (S4C Fig) [18]. Consistent with this, we also observed that mitochondrial morphology is only clustered in *clu1Δ* cells under respiratory conditions (S4D and S4E Fig).

*Drosophila* Clu has been shown to localise near mitochondria [12] (Fig 3F). To further clarify the subcellular localisation of Clu foci, we employed correlative light and electron microscopy (CLEM). In 'refed' *Drosophila* egg chambers, where GFP-Clu foci are abundant, we found that Clu foci consistently localised adjacent to mitochondria, not directly overlapping (Fig 3G). Notably, we did not observe any organelle or distinct structure between the Clu foci and mitochondria, suggesting that they are likely in direct contact. Moreover, GFP-Clu foci did not appear to be bound by any detectable membrane, indicating they form a membraneless organelle (Fig 3G).

The biophysical properties of membraneless organelles have been intensely characterised recently and have been found to form by liquid-liquid phase separation (LLPS) driven by weak and multivalent interactions [26]. To explore whether Clu/Clu1 foci are formed by LLPS, we treated ovaries from fed flies and PD yeast cells exhibiting numerous Clu/Clu1 foci with 1,6-hexanediol, an aliphatic alcohol which disassembles LLPS condensates by disrupting weak hydrophobic interactions [27–30]. Live imaging of GFP-Clu in egg chambers treated with 1,6-hexanediol showed a steady dissolution of the GFP-Clu foci over time in contrast to untreated egg chambers (Fig 4A). Complementing this, Clu1-GFP yeast cells were permeabilised with digitonin and incubated with increasing concentrations of 1,6-hexanediol. With 5% incubation, the number and size of Clu1 foci decreased, while with 10%, the foci were completely dissolved (Fig 4B).

The use of 1,6-hexanediol to investigate LLPS has some caveats [31,32], so we sought an orthogonal approach to analyse the dynamic nature of these foci. We used fluorescence recovery after photobleaching (FRAP) to analyse Clu1-GFP foci in PD cells and found that, indeed, Clu1 exhibited partial mobility into these structures (Fig 4C). The fluorescence recovery began immediately after bleaching and continued to increase, recovering by approximately 50% over the 600 s recording time-frame (Fig 4D). These results show that a substantial fraction of Clu1 in foci is dynamic and exchanges with cytoplasm, despite Clu1's abundance in the cytoplasm being low. Taken together, the preceding results support that Clu/Clu1 foci are membraneless, dynamic and form by LLPS; thus, hereafter, we refer to them as Clu/Clu1 granules.

## Clu granules contain RNA and require mRNAs engaged in translation

Many biomolecular condensates formed by RNA-binding proteins contain RNAs and rely on RNAs for their formation [33–35]. As evidence indicates that Clu/Clu1 are RNA-binding proteins [16], we hypothesised that Clu granules contain RNAs, which may influence Clu granule stability. To test this, we first performed oligo(dT) fluorescence in situ hybridisation (FISH) on egg chambers from refed flies when GFP-Clu granules are abundant. As expected, poly(A) RNAs were abundant and widely distributed in egg chambers confounding localisation analysis (Fig 5A). However, we reasoned that mild RNase A treatment of fixed samples would readily degrade cytoplasmic mRNAs while sparing any located within Clu granules. Indeed, following RNase treatment when most poly(A) signal was lost from the cytoplasm, some signal was still detectable co-localising with Clu granules (Fig 5A and 5B). To address the potential impact of RNAs on granule stability, we next assessed the impact of RNase treatment on live egg chambers from refed flies. Here, RNase treatment resulted in the complete disassembly of most Clu granules (Fig 5C and 5D), similar to the effect on PBs [36]. Together, these results indicate that Clu granules contain mRNAs which play a crucial role in the formation and/or stability of the granules.

Since the formation of SGs and PBs depends on mRNAs released from ribosomes, we sought to determine whether the formation of *Drosophila* Clu granules depends on a similar mechanism. In yeast, flies and mammalian cells, pre-treatment with cycloheximide (CHX) – an inhibitor of protein synthesis that traps mRNAs in polysomes – prevents the formation of SGs and PBs upon stress exposure [19,36–38]. To test this, we took advantage of the fact that insulin can

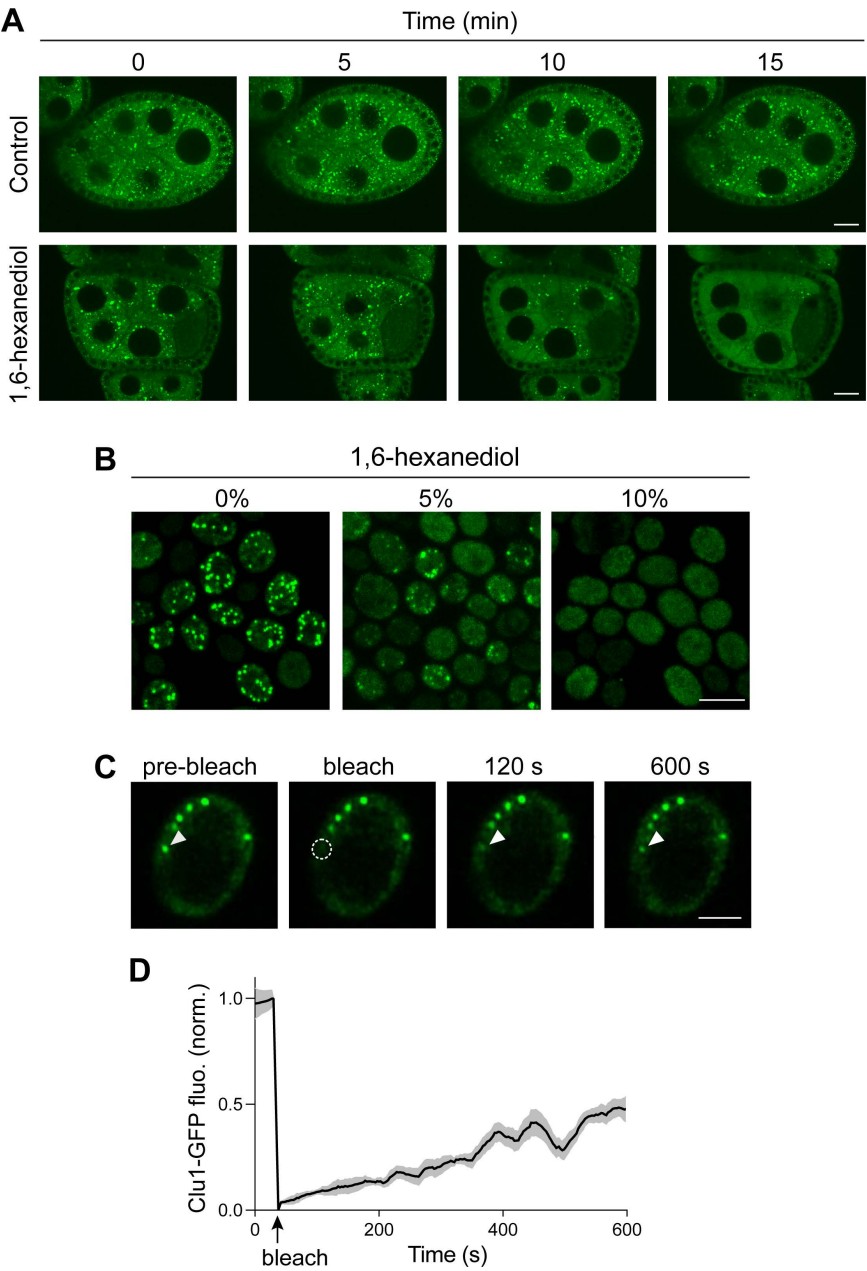

**Fig 4. Clu/Clu1 foci have liquid-like properties.** (A) Representative live imaging of a GFP-Clu egg chamber incubated in Schneider's media containing insulin, treated with 1,6-hexanediol or vehicle and imaged over a 15 min period. (B) Images of PD Clu1-GFP cells fixed after being permeabilised with digitonin for 2 min, followed by 5 min treatment with 0, 5 and 10% 1,6-hexanediol. (C) Representative fluorescence-recovery after photobleaching (FRAP) images of a Clu1-GFP foci in a PD cell. Arrowheads indicate the bleached Clu1-GFP foci, and the dashed circle represents the photobleached area. (D) Normalised fluorescence recovery over time (mean±SEM; n = 6 cells from 3 independent experiments). Scale bars: A = 20 µm; B = 5 µm; C = 2 µm.

trigger granule formation in dissected egg chambers (S2 Video) [18]. In contrast to SGs and PBs, pre-treatment with CHX did not prevent the formation of Clu granules upon addition of insulin (Fig 5E), indicating that Clu granules do not form with mRNAs that exit translation. However, this did not exclude the possibility that Clu granules may form with free mRNAs that have not yet engaged in translation. Therefore, we next assessed the effect of puromycin, which dissociates

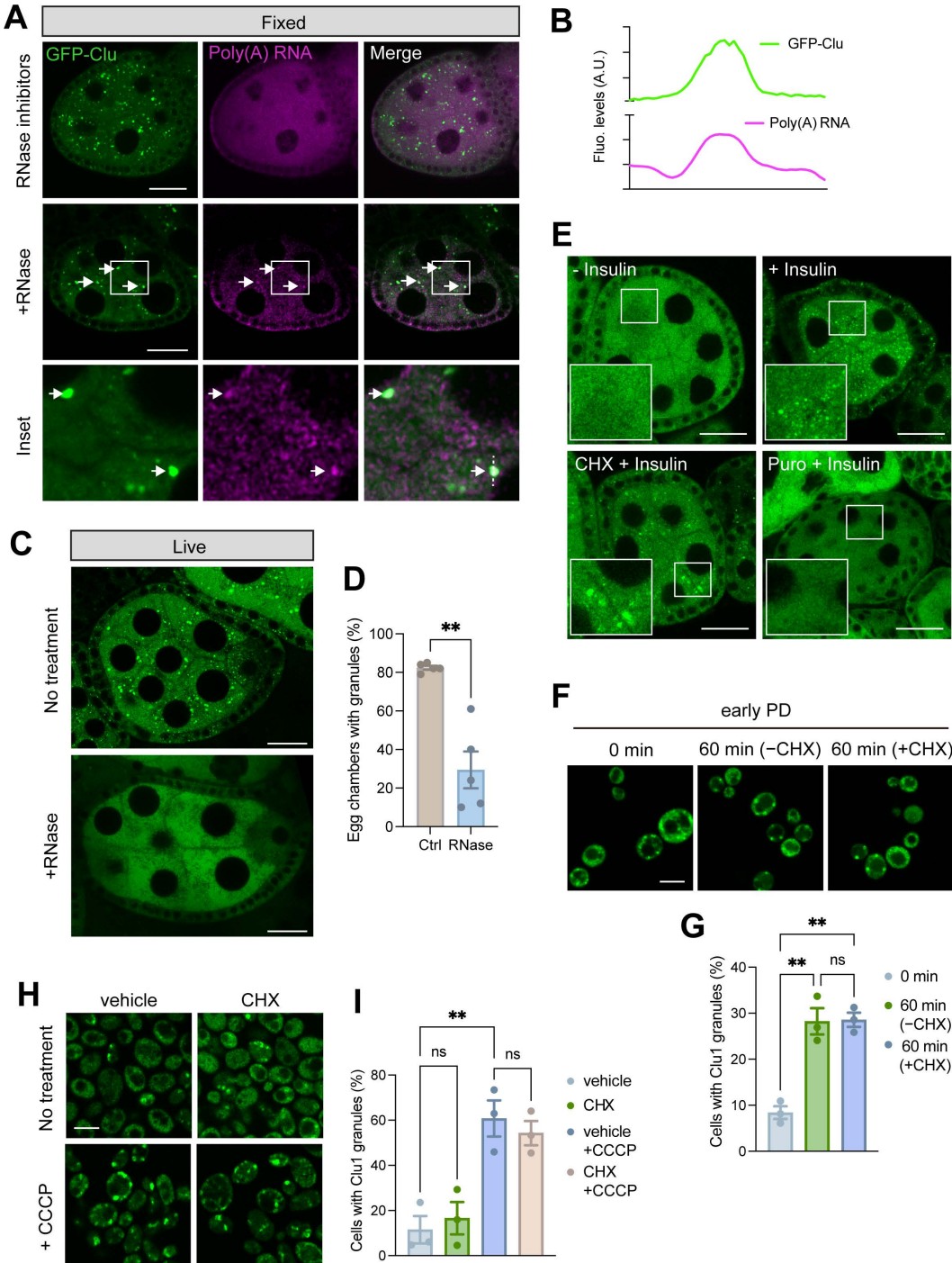

**Fig 5. Clu/Clu1 granules are RNase- and puromycin-sensitive but cycloheximide-insensitive.** (A) Fluorescence in situ hybridisation (FISH) of poly(A) RNAs using an oligo-dT(45)-Alexa647 in fixed egg chambers treated with RNase inhibitors or RNase A. The inset highlights two GFP-Clu granules enriched in poly(A) RNAs. (B) Intensity plot corresponding to the dashed line drawn in A (inset). (C) Dissected live egg chambers from GFP-Clu flies permeabilised and incubated with or without RNase A for 5 min and analysed by confocal microscopy. (D) Quantification of the percentage of egg chambers containing Clu granules per fly shown in C (mean ± SEM; n = 5 animals; unpaired t-test with Welch's correction; ** P < 0.01). (E) Confocal images of egg chambers subjected to different treatments after being dissected in Schneider's media: a control with no additional treatment (40 min) (- Insulin), incubation with insulin (30 mins) 10 min after dissection (+ Insulin), or pre-treatment with CHX (CHX + Insulin) or puromycin (Puro + Insulin) for 10 min

followed by incubation with insulin (30 mins). (F) Confocal images and (G) quantification of Clu1-GFP cells in the early PD phase and treated with or without CHX for 60 min. (H) Confocal images and (I) quantification of Clu1-GFP cells with granules in cells shifted to galactose-containing media for 6 h, pre-treated with CHX or vehicle for 10 min and then followed by CCCP or vehicle treatment for 15 min. Charts in G and I are presented as mean ± SEM; n = 3 biological replicates; > 100 cells each biological replicate; one-way ANOVA, ns = non-significant, ** P < 0.01. Scale bars: A, C, E = 20 μm; F, H = 5 μm.

ribosomes from the mRNA by releasing truncated polypeptides [39]. In contrast to CHX, pre-treatment with puromycin before insulin addition prevented the formation of Clu granules (Fig 5E), indicating that Clu granule formation depends on mRNAs being actively engaged with the ribosome.

To investigate whether yeast Clu1 behaves like fly Clu, we incubated Clu1-GFP cells in early PD phase with CHX and quantified Clu1 granule formation. Similar to our observation in fly egg chambers, the addition of CHX did not affect the formation of Clu1 granules compared to the control (Fig 5F and 5G). Notably, pre-treatment with CHX also did not prevent the formation of Clu1 foci in mitochondrial stress conditions (respiring cells treated with CCCP) (Fig 5H and 5I), showing again that Clu1 granules formed during a gradual shift to respiration or by acute mitochondrial stress are similar in nature.

Taken together, these results indicate that Clu/Clu1 granule formation depends on mRNAs being engaged in translation, but not on the release of mRNAs from ribosomes, highlighting a unique aspect of their assembly mechanism compared to other RNA-containing granules like SGs and PBs.

## Yeast Clu1 regulates the translation of nuclear-encoded mitochondrial proteins

To further characterise the role of Clu1 in yeast, we performed a proximity-dependent biotinylation assay – BioID [40,41] – to identify potential direct or indirect Clu1 interactors. We generated a knock-in *CLU1-GFP-BirA** strain and a strain expressing cytosolic BirA* as control and validated their correct expression and localisation (S5 Fig). Mass spectrometry identified 26 'Clu1-proximal' proteins in the log phase and 22 in the PD phase, 4 of which are common between the two phases (Fig 6A, and S1 Table). Notably, 12 proteins detected in the log phase are nuclear-encoded mitochondrial proteins, most of which localise in the mitochondrial matrix and are involved in metabolic processes (Fig 6A and S1 Table). These include Pda1 and Lat1 (pyruvate dehydrogenase complex), Aco1 (Aconitase; TCA cycle), and the aldehyde dehydrogenases Ald4 and Ald5 (acetate production). Additionally, we detected three cytosolic ribosomal proteins and the translation initiation factor eIF-4B (Tif3). In the PD phase, we also identified nuclear-encoded mitochondrial proteins, including Aco1 and Lat1, which were common to both phases, five ribosomal proteins (RpL20a being common between phases), and the translation initiation factor eIF5 (Tif5). Additionally, we detected the RNA-binding protein Scp160 and the tRNA synthetase Ses1, both showing high co-expression scores with Clu1 according to the SPELL database [42] (Scp160: 3.6, most correlated gene; Ses1: 3.1).

To validate interactions identified by BioID *in vivo*, we used the bimolecular fluorescence complementation assay (BiFC) [43,44]. We tested Rpl17b, Aco1, Scp160, Tif3 and Tif5 and two control non-interactors (Sod2 and Pyc2), each fused with the C-terminal fragment of Venus (VC) at their genomic C-termini [45] and co-expressed with Clu1 tagged with the Venus N-terminal fragment (VN) [46]. Reconstituted Venus fluorescence confirmed the interaction of Clu1 with the selected BioID hits in fermenting (glucose) and respiring (glycerol) conditions, while controls showed no signal (Fig 6B). In glucose media, the Venus signal was diffuse, except in cells co-expressing Clu1-VN and Aco1-VC, which displayed a few small puncta (Fig 6B). In glycerol, Venus localisation changed from diffuse to punctate in cells co-expressing Clu1-VN and Rpl17b-VC, Scp160-VC, Tif3-VC, or Tif5-VC, and puncta size and number increased in the Clu1-VN Aco1-VC strain (Fig 6B). Co-expression with a mitochondrial marker (mito-BFP) or Clu1-mCherry showed that Venus puncta localised near mitochondria (Fig 6C) and co-localised with Clu1-mCherry granules in respiratory conditions (Fig 6D), confirming these interactions happen in granules. These results indicate that Clu1 co-localises with cytosolic ribosomal and nuclear-encoded mitochondrial proteins, likely before their mitochondrial import.

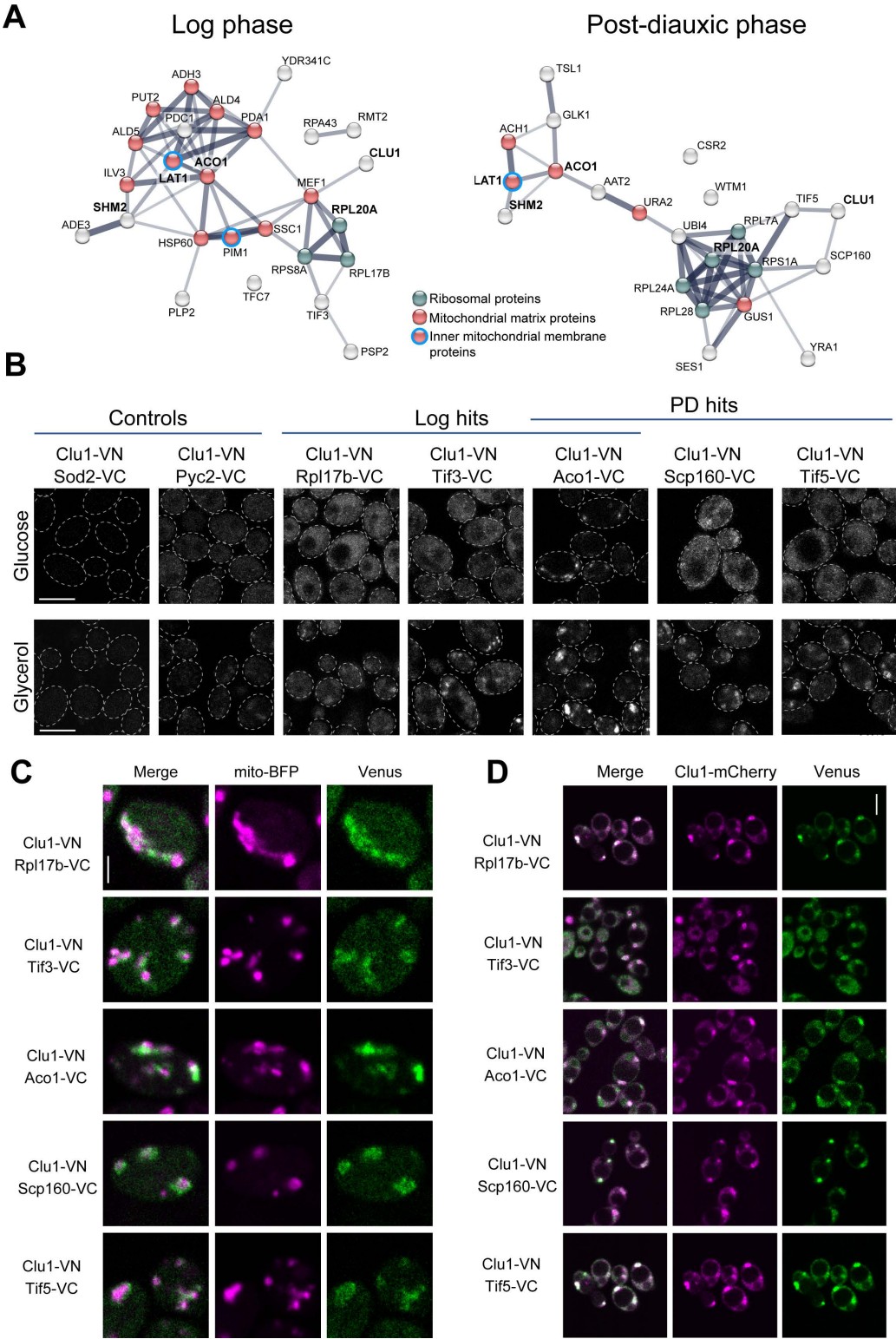

**Fig 6. Clu1 is in a complex with ribosomes and mitochondrial proteins.** (A) STRING association network analysis (based on physical and functional interactions) of yeast Clu1 interactors identified by BioID in the log or PD phases. Highlighted in bold are proteins found in common in both growth

phases. (B) Confocal microscopy analysis of bimolecular fluorescence complementation (BiFC) assay of cells co-expressing Clu1 fused with the Venus N-terminal fragment (Clu1-VN) and either non-interactor controls (Sod2 and Pyc2) or BioID hits (Rpl17b, Tif3, Aco1, Scp160, and Tif5) fused to the Venus C-terminal fragment (protein-VC). Cells were either grown in glucose media until mid-log phase (upper panels) or shifted to glycerol media for 24 h (lower panels). (C, D) Localisation of where Clu1-BioID hits interact visualised by the Venus reconstitution (green) relative to mitochondrial marker mito-BFP (magenta) (C) and Clu1-mCherry (magenta) (D) in cells shifted to glycerol for 24 h. Scale bars: B, D = 5 μm; C = 2 μm.

The proximity of Clu1 to ribosomal and nuclear-encoded mitochondrial proteins led us to investigate whether Clu1 regulates translation by analysing the translatome of *clu1Δ* cells compared to the wild-type strain. To achieve this, we used puromycin-associated nascent chain proteomics (Punch-P), which consists of isolating intact polysomes that will incorporate biotinylated puromycin into nascent polypeptides, giving a snapshot of active translation [47,48]. Resulting biotinylated polypeptides from wild-type and *clu1Δ* polysomes were purified using streptavidin and identified by mass spectrometry (Fig 7A).

In respiring cells, we detected 360 proteins, 47 of which were significantly different between wild-type and *clu1Δ* strains (Fig 7B and S2 Table). Of these 47 proteins, 21 were less translated in the absence of Clu1 (Fig 7B). Notably, 6 out of the 21 proteins, such as Aco1 and Pda1, were also identified in the BioID assay (Fig 6A), further supporting that Clu1 regulates their translation. Most proteins with reduced translation in *clu1Δ* were nuclear-encoded mitochondrial proteins, mainly metabolic proteins involved in the TCA cycle, pyruvate metabolism and oxidative phosphorylation (Fig 7C). This likely contributes to the *clu1Δ* strain deficient growth in ethanol-containing media (S4C Fig). Proteins more translated in *clu1Δ* were cytosolic proteins and included ribosomal and metabolic proteins involved in glycolysis/gluconeogenesis or amino acid biosynthesis (Fig 7D).

Less translated proteins also included several chaperones (Fig 7C), leading us to hypothesise that *clu1Δ* cells may be more sensitive to heat-shock stress. To test this, we subjected both wild-type and *clu1Δ* cells to acute heat shock under three conditions: log phase in glucose-containing media, lag phase after transition to ethanol media (adapting to respiration), and log phase in ethanol media (adapted to respiration). Interestingly, *clu1Δ* cells were specifically more sensitive to heat shock during adaptation to respiration (S6 Fig), a phase characterised by extensive mitochondrial protein translation [49]. This sensitivity may result from increased demand for chaperones during this phase, which, combined with their reduced levels in *clu1Δ* cells, likely results in impaired mitochondrial protein import and folding.

The effects on translation caused by loss of Clu1 coupled with Clu1 binding to ribosome proteins led us to hypothesise that translation efficiency (TE) was affected in *clu1Δ*, rather than simple variations in steady-state mRNA levels. Thus, we calculated the TE of proteins with altered translation by determining the ratio of nascent protein levels (Punch-P) to their corresponding steady-state mRNA levels. For most proteins with reduced translation, their TE was lower (Figs 7E and S7A), while the majority of proteins with increased translation showed higher efficiency (Figs 7E and S7B). Quantification of mRNA levels from the same polysome samples used for Punch-P showed no differences in mRNA levels for less translated proteins in *clu1Δ* (Aco1, Pda1, Hsp60, Ssc1, Atp1) (Fig 7F), suggesting that Clu1 acts at the ribosome level to regulate their translation. In contrast, mRNA levels were elevated for the more translated proteins (Pgk1, Leu1, Ade3) (Fig 7F), with their increased translation possibly reflecting a compensatory response. Altogether, these findings suggest that Clu1 plays a critical role in coordinating mitochondrial protein translation, which is essential for metabolic adaptation and stress response.

## Clu1 interacts with nuclear-encoded mitochondrial mRNAs, via ribosome interaction

Our findings suggested that yeast Clu1 regulates the translation of mRNAs of nuclear-encoded mitochondrial proteins, similar to its mammalian orthologue, CLUH [13]. Since CLUH is known to directly bind to mRNAs as an RNA-binding protein [13], we explored whether Clu1 functions in a similar way. We also sought to test the hypothesis that Clu1 interacts with mRNAs during translation. For this, we performed RNA immunopurification (RIP) of Clu1-GFP and GFP (control) under conditions where ribosomes were kept intact but stalled by CHX, and compared with conditions where ribosomes

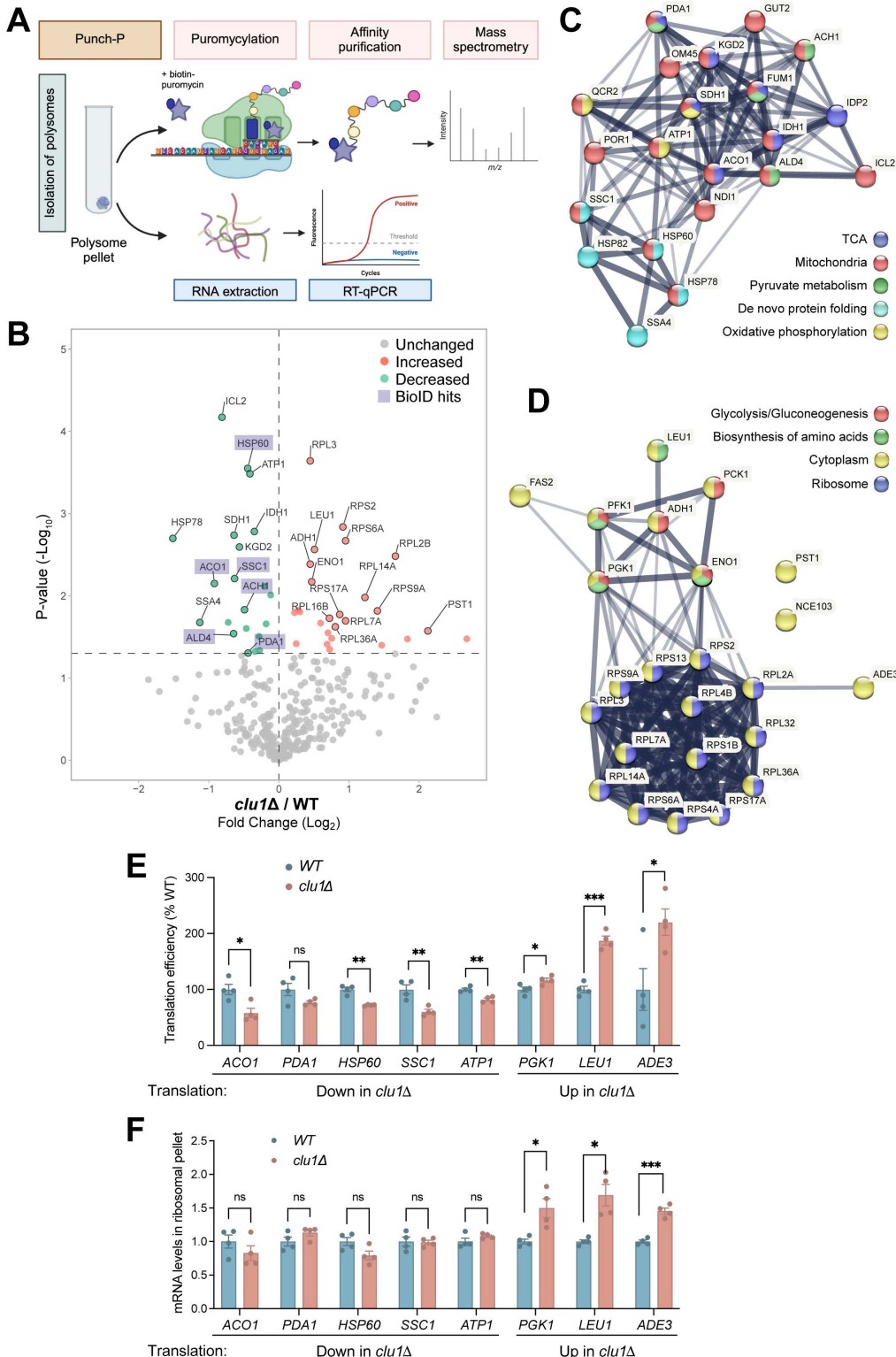

**Fig 7. Translation of specific nuclear-encoded mitochondrial proteins is affected by the absence of Clu1.** (A) Experimental workflow schematic of the Punch-P assay in which biotinylated-puromycin is incorporated into nascent proteins via actively translating polysomes, followed by purification and identification of the newly synthesised proteins by mass spectrometry. Polysomes isolated via a sucrose cushion ultracentrifugation were used either for

the Punch-P assay or RNA extraction for RT-qPCR analysis. Image created with BioRender. (B) Volcano plot of proteins detected in the Punch-P assay comparing control (WT) with *clu1Δ* cells. The X-axis represents the fold change in protein translation and the Y-axis corresponds to the P-value. Shaded hits correspond to proteins found in proximity to Clu1 (BioID hits). (C, D) STRING interaction network of the proteins with reduced (C) or increased (D) translation in the *clu1Δ* cells compared to control. (E) Translation efficiency (TE) quantification of proteins with altered translation in the *clu1Δ*. TE was calculated as the ratio of nascent protein levels (Punch-P) to the steady-state levels of their corresponding mRNA, and normalised to the wild-type levels (mean ± SEM; n = 4 samples; unpaired t-test; * $P < 0.05$, ** $P < 0.01$, *** $P < 0.001$). (F) RT-qPCR analysis of mRNAs of selected hits that showed decreased or increased translation in the Punch-P assay in control or *clu1Δ* cells. mRNA levels were normalised to 25S rRNA (mean ± SEM; n = 4 samples; unpaired t-test; * $P < 0.05$, *** $P < 0.001$).

were dissociated by EDTA. We predicted that transcripts whose translation was altered in the absence of Clu1 (Punch-P assay) and/or encode proteins found in proximity to Clu1 (BioID assay) would likely physically interact with Clu1. Thus, we selected *ACO1, PDA1, HSP60, SSC1, ATP1* and *ADH3* to quantify by RT-qPCR. As negative controls, we selected transcripts encoding two nuclear-encoded mitochondrial proteins not predicted to interact with Clu1 (*SOD2* and *ATP2*). Indeed, in intact ribosomes the mRNAs that we predicted to interact with Clu1 were highly enriched in the Clu1-GFP RIP compared to the GFP control, while the negative controls showed no enrichment (Fig 8A). Surprisingly, given the known role of CLUH to directly bind mRNAs [13], this enrichment was completely lost when ribosomes were disrupted by EDTA (Fig 8B). Since this was contrary to our expectations of Clu1 being an RNA-binding protein, and since EDTA could have effects besides disrupting ribosomes, we used puromycin treatment as an alternative method to disrupt ribosome-RNA interactions. Similar to EDTA, Clu1 target binding was significantly lost with puromycin (Fig 8B).

In parallel, we found both *18S* and *25S* rRNAs enriched in the Clu1-GFP RIP when ribosomes were intact. This enrichment was also lost upon dissociation of ribosomes (Fig 8C). We further corroborated the association of Clu1 with the ribosome by co-immunoprecipitating Clu1 and the ribosomal protein Rpl3 (Fig 8D). These results support that Clu1 forms a complex with ribosomes and interacts with specific nuclear-encoded mitochondrial mRNAs only when they are actively engaged in translation.

To further explore the association of Clu1 with ribosomes and its potential changes under varying metabolic conditions, we performed sucrose gradient fractionation and assessed its co-migration with ribosomes. In addition, we questioned whether the Clu1 interaction with the ribosome was mRNA-dependent. For this, we analysed cells in the log phase under three different conditions: with CHX to maintain the integrity of monosomes and polysomes; with CHX followed by RNase treatment that converts polysomes into monosomes; and with EDTA to disassemble polysomes and monosomes. The ribosome profile was monitored by 254 nm absorbance and by immunoblotting Rpl3 for the large ribosomal subunit. As a control, we immunoblotted for Pgk1, a glycolytic monomeric enzyme not involved in ribosomal or high-order protein complexes [50]. When polysomes and monosomes were preserved (CHX), Clu1 was abundant in the light fractions, corresponding to free Clu1 or Clu1 binding to small proteins or RNAs, but was also present in the high-density fractions co-migrating with monosomes and polysomes (Fig 8E). Upon treating the lysate with RNase, as expected, polysomes were disrupted and ribosomes accumulated in the monosome fractions. Here, Clu1 disappeared from the heaviest fractions and became enriched in the monosome fractions (Fig 8F), indicating that the interaction with ribosomes is mRNA-independent. Upon disassembling ribosomes with EDTA, Clu1 completely disappeared from the high-density fractions, as did ribosomes, further confirming its association with the ribosome (Fig 8G). The similarity between Clu1 and ribosome migration profiles in all conditions tested in the log phase indicates that Clu1 binds ribosomes, either in polysome or monosome form, independently of the interaction with mRNA.

To further study Clu1 in the context of ribosomes, we chemically cross-linked Clu1-GFP log cells to stabilise weaker interactions that might dissociate during the long sucrose gradient ultracentrifugation procedure. Cross-linking significantly shifted the migration pattern of Clu1, from the lighter fractions to now being highly enriched in the polysome fractions (Fig 8H), indicating that the interaction between Clu1 and ribosomes may be transient and/or weak. To further confirm that the presence of Clu1 in the polysome fraction is due to its interaction with the ribosome, we treated the cross-linked lysate

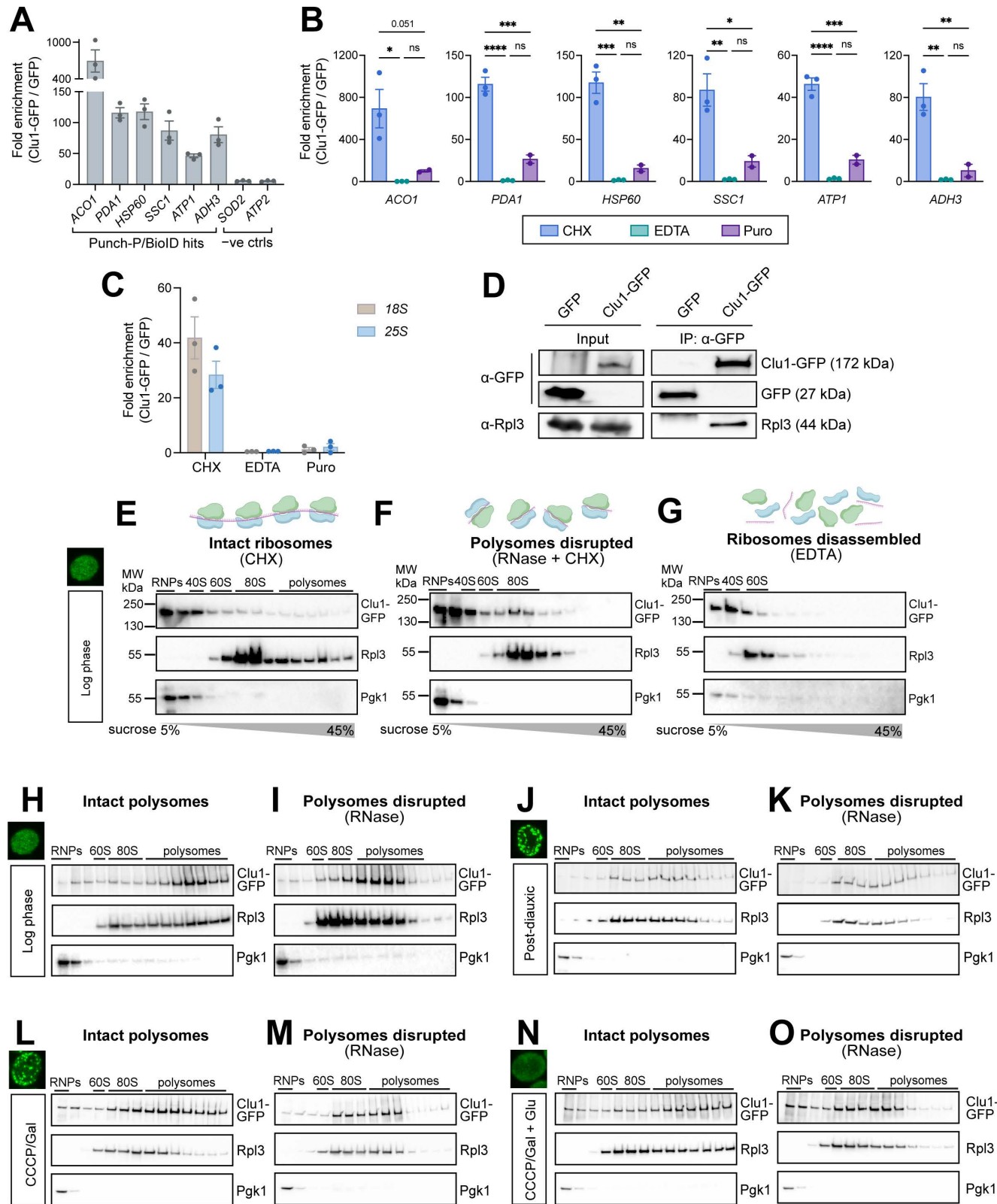

**Fig 8. Clu1 interacts with translating nuclear-encoded mitochondrial mRNAs.** (A, B) Clu1-GFP RNA immunoprecipitation (RIP) was performed under (A) ribosome-preserving (CHX) or (B) ribosome-disassembling conditions (EDTA or puromycin (Puro)), using GFP-expressing cells as a control.

RNA was extracted following RIP and mRNA levels of BioID and/or Punch-P hits (*ACO1*, *PDA1*, *HSP60*, *SSC1*, *ATP1* and *ADH3*) were quantified by RT-qPCR, alongside non-interactor controls (*ATP2*, *SOD2*). The Y-axis shows the relative enrichment of Clu1-GFP RIP versus the GFP control. The same analyses for CHX samples are shown in A and B for comparison (mean±SEM, n = 3; one-way ANOVA with Tukey's multiple comparison test; * $P < 0.05$, ** $P < 0.01$, *** $P < 0.001$, **** $P < 0.0001$, ns = non-significant). (C) RT-qPCR analysis of RIP samples as in A and B for *18S* and *25S* rRNA (mean±SEM, n = 3). (D) Immunoblot of GFP immunoprecipitation (IP) of GFP or Clu1-GFP expressing cells, detecting physical interaction with the ribosomal protein Rpl3. (E-G) Sucrose continuous-density gradients and ribosome profiling performed under polysome-preserving (CHX), polysome-disrupting (RNase + CHX) or ribosome-disassembling (EDTA) conditions of log phase grown cells. Representative immunoblots of the co-sedimentation of Clu1-GFP with ribosomes (Rpl3). Pgk1 serves as an indicator of cytosolic/soluble protein sedimentation. Blots are representative of 3 independent replicates. (H-O) Sucrose density gradients and ribosome profiling, similar to E-G, performed following in vivo formaldehyde cross-linking to preserve interactions under different growth conditions: log phase (H, I) or PD phase (J, K); or metabolic/stress conditions: CCCP/Gal (galactose) (L, M) or CCCP/Gal + Glu (glucose) (N, O). (I, K, M, O) Sucrose gradients of same lysates used for H, J, L, N, treated with RNase to disrupt polysomes. Blots are representative of 3 independent replicates. Thumbnail images are to indicate presence or absence of Clu1 granules in the indicated growth conditions. Illustrations were created using BioRender.

with RNase (Fig 8I). Although this treatment had a modest effect in disrupting polysomes, as observed in the non-cross-linked cells, Clu1 clearly followed the ribosomes' migration pattern, indicating that Clu1 migrates into the denser fractions of the gradient via interaction with ribosomes.

To determine whether the association of Clu1 with the ribosome changes upon metabolic transitions or mitochondrial stress when Clu1 is in granules, we also analysed cross-linked cells in PD phase or under respiratory stress (Galactose + CCCP). In both conditions, the abundance of high-density polysomes decreased, indicating a reduced general translation, and notably, although some Clu1 shifted towards the monosome fractions, it was still highly abundant in polysome fractions (Fig 8J and 8K). We further assessed the Clu1–ribosome interaction when granules disappear upon shifting the metabolism towards fermentation, as we observed when spiking glucose into the media of cells grown in respiratory media and then stressed with CCCP (Galactose + CCCP) (see Fig 2C). Upon brief (15 min) incubation with glucose, both ribosomes and Clu1 moved to the heavy polysomes fractions (Fig 8N) when compared to cells before glucose addition (Fig 8L). Altogether, these results show that Clu1 interacts with polysomes independently of the metabolic condition and its localisation.

As previously, to confirm that this co-migration is due to an mRNA-independent association of Clu1 with ribosomes, we treated these lysates with RNase and observed that the Clu1 migration profile completely mirrored the ribosome profile in all conditions tested (Fig 8K, 8M and 8O). Altogether, these results show that the majority of Clu1 is always associated with ribosomes, independently of the mRNA interaction, whether Clu1 is diffuse or in granules, and independently of whether the metabolic shift is gradual or acute.

## Discussion

Mitochondria are key organelles for cellular adaptation to metabolic shifts and stress. Post-transcriptional regulation of their proteome plays a crucial role in how mitochondria respond to various conditions. The *Drosophila* protein Clu and its mammalian orthologue CLUH were shown to be involved in regulating translation and maintaining mitochondrial function [11–17,51–54], but their precise roles are poorly understood.

In this study, we provide novel insights into the function of the *Drosophila* Clu and the yeast orthologue Clu1. Both proteins dynamically change their localisation upon metabolic transitions, shifting from a diffuse cytosolic distribution to granule formations adjacent to mitochondria, suggesting a regulatory mechanism in mitochondrial metabolism. The dynamic distribution of fly Clu triggered by fasting and refeeding or by the addition of insulin to cultured egg chambers corroborates earlier observations [18]. Egg chambers are cellular structures highly sensitive to nutritional changes [55], however, the exact metabolic changes that occur during these rapid nutritional fluctuations, or insulin addition, are poorly characterised. Nevertheless, Clu granules appear in a transition from low to high metabolic conditions, which likely translates into higher respiratory metabolism [56]. In yeast, we more precisely defined the metabolic trigger for the formation of Clu1 granules

as the transition from fermentation to respiration – a process involving profound transcriptional, translational and metabolic changes, particularly at the mitochondrial level [49]. Our findings are in line with a recent study demonstrating that CLUH forms granules in primary mouse hepatocytes when cells are switched from nutrient-rich to nutrient-poor media, indicating that granule formation in response to metabolic changes is a conserved feature [17].

Membraneless organelles are cellular compartments that, despite lacking a lipid membrane, play important roles in organizing the cellular content. When dysregulated, they may contribute to the pathogenesis of diseases such as amyotrophic lateral sclerosis [35,57]. These organelles are formed through liquid-liquid phase separation (LLPS), driven by weak and transient interactions between multivalent molecules, including proteins and, in many cases, mRNAs [26,57]. Their rapid assembly and disassembly along with easy exchange of components with the external environment allows cells to quickly adapt to metabolic changes and various stresses. Previous studies had observed that Clu1/Clu/CLUH proteins form foci, but definitive evidence that these are biomolecular condensates, including confirming the absence of a membrane, was lacking [17,18]. Despite several similarities with PBs and SGs, such as being rapidly dynamic and reversible, Clu1/Clu granules are distinct structures, not co-localising with classic markers of these organelles and not forming under most typical stress conditions. This shows functional similarities to the mammalian CLUH granules, which were also shown to be independent of SGs despite co-localising with typical SG component G3BP1 [17]. Using a CLEM approach, we found that fly Clu granules are not delimited by a membrane and are in close proximity to mitochondria. In addition, we observed that Clu/Clu1 granules quickly dissolve upon treatment with 1,6-hexanediol, which, despite some caveats related to its use [31,32], is consistent with these granules being formed by LLPS [27–30]. Interestingly, Clu1 exhibits moderate dynamics in granules, with ~50% fluorescence recovery over 10 min. This is faster than some stable, gel-like assemblies such as Balbiani bodies based on the dynamics of its component Xvelo-GFP, which recovers only ~20% over 1 h [58]. However, it is slower than highly dynamic, liquid-like condensates such as PBs, where certain components (for example, Sbp1 and Eap1) recover nearly 100% within 150 s [59]. Interestingly, other PB components, such as Dcp2 and Edc3, show little to no measurable recoveries, highlighting that proteins within the same granule can have markedly different mobilities. In future studies, it will be interesting to assess the behaviour of other proteins in Clu1 granules to better understand their material properties and internal organisation.

Building on recent observations of Clu/CLUH dynamic localisation, we focussed our investigation on the nature and formation of the Clu/Clu1 granules, including their components. We first found that Clu granules contain RNA and require it for stability. Interestingly, Clu granule formation was prevented by dissociating ribosomes with puromycin, indicating that granules require mRNAs engaged in translation to be formed. Our results corroborate a study in primary mouse hepatocytes that suggested a subset of CLUH granules contain translating ribosomes [17]. In contrast, stalling translation with CHX had no impact on granule formation in flies or yeast, also consistent with observations for CLUH in HeLa cells [17] and in contrast to SGs and PBs [19,36–38]. Altogether, these results indicate that Clu/Clu1 granule formation is tightly linked to ribosome-associated mRNAs.

Our data strongly implicates a role for yeast Clu1 in regulating the translation of specific nuclear-encoded mitochondrial mRNAs. First, the characterisation of Clu1 interactome by BioID showed mainly ribosomal proteins, nuclear-encoded mitochondrial proteins, RNA-binding proteins and translation initiation factors. An early study has suggested that Clu1 interacts with the eukaryotic translation initiation factor 3 (eIF3) [60], but we did not find evidence of this interaction. In line with our findings, a recent BioID analysis of CLUH from a human colorectal carcinoma cell line revealed a similar interactome [61]. In addition, Clu1, Clu, CLUH and the *Arabidopsis thaliana* orthologue, Friendly mitochondria (FMT) [62], were shown to interact with ribosomal proteins, highlighting the conservation of this interaction mechanism [16,51]. Interestingly, when characterising the interaction of Clu1 with a subset of the interacting proteins (Scp160, Tif3, Tif5, Rpl17, and Aco1) by BiFC, we found that these interactions occur regardless of Clu1 condensation state and metabolic state, suggesting that Clu1's role is independent of its condensation state. Our translatome data further supports the role of Clu1 in regulating the translation of specific mitochondrial proteins. We found that the absence of Clu1 decreases the efficiency of

translation of various nuclear-encoded mitochondrial proteins, many of which we identified as being in proximity to Clu1 by BioID. It remains to be determined whether this decrease in translation efficiency is caused by a reduced number of ribosomes per polysome (indicating a possible decrease in translation initiation) or by slower elongation rates.

Additionally, we found that mRNAs encoding proteins proximal to Clu1 and/or dysregulated in its absence, such as *ACO1, PDA1* and *HSP60*, physically interact with Clu1. Notably, the orthologous mammalian mRNAs were also shown to interact with CLUH [13], revealing a remarkable conservation of the protein-RNA interaction and functional homology between these proteins.

Surprisingly, our Clu1 RIP findings show that Clu1 interacts with mRNAs only when monosomes and polysomes are intact, supporting that Clu1 only interacts with mRNAs engaged in translation. Furthermore, our sucrose gradient analysis showed that Clu1 associates with ribosomes independently of Clu1 interacting with mRNA. Together these results indicate that Clu1's interaction with mRNAs depends on their translation and involves a direct interaction with the ribosome. The striking similarity between Clu1- and CLUH-interacting mRNAs further supports the conserved function between the yeast and human proteins. As CLUH was shown to directly bind mRNAs [13,51,61] and to interact with ribosomal proteins [51,61], we propose that Clu1 and CLUH interact simultaneously with the ribosome and mRNAs, with this interaction occurring specifically when mRNAs are being translated, likely by influencing their localisation, stability, or translation efficiency. Importantly, we found that Clu1 associates with ribosomes when they are in polysomes, likely while actively translating, both when it is diffuse and when in granules, whether triggered by a switch to respiration or mitochondrial stress. The prevention of Clu granule formation by dissociating ribosomes with puromycin suggests that the presence of ribosomes is crucial for this process. Altogether, these findings support the idea that Clu1/Clu granules contain translating ribosomes. Since membraneless organelles can facilitate or limit interactions of their components, by concentrating interacting molecules or isolating specific macromolecules from the surroundings [63], we hypothesise that Clu/Clu1 granules may promote or slow down the translation of these proteins even in adverse conditions where perhaps the mRNA targets of Clu1/Clu would be degraded.

Further research is necessary to fully elucidate the complexity of these mechanisms, in particular, how Clu1 (and, by extension, Clu and CLUH) recognise changes in metabolic conditions to mediate molecular changes. We speculate that this could be by posttranslational modifications, for instance, via other metabolic sensors, the differing presence/absence of binding partners, or an as yet unidentified mechanism. Nevertheless, our findings define Clu/Clu1 granules as unique membraneless organelles that likely regulate the production of mitochondrial proteins near the mitochondria. This has significant implications for understanding the dynamic and homeostatic processes regulating mitochondrial function. It will be interesting to investigate in the future if the unique subcellular environment within the granules has an impact in the regulation of translation of these transcripts.

## Materials and methods

### *Drosophila* methods

**Stocks and husbandry.** *Drosophila melanogaster* flies were raised under standard conditions in a temperature-controlled incubator with a 12 h:12 h light:dark cycle at 25 °C and 65% relative humidity, on food consisting of malt extract, molasses, cornmeal, yeast, agar, soya powder, water, propionic acid, nipagin. GFP-Clu trap line (clu[CA06604]) was obtained from Rachel Cox and Tral-RFP was kindly provided by Daniel St Johnston [64].

**Fed and refed time course.** Crosses were set up with 20 GFP-Clu females and 10 males (one-day-old flies) in bottles with yeast paste for one day. Flies were starved overnight in tubes with tissue paper wet in water only. Flies were refed with yeast paste containing bromophenol blue (to control for feeding) for up to 6 h and then starved again in wet paper for up to 1 h and a half. Female ovaries were dissected in different intervals on Grace's media (Thermo Fisher Scientific; 11605045) when flies were being fed or PBS when starved and then fixed in 4% formaldehyde (Thermo Fisher Scientific; 28908) made in PBS containing 0.3% of Triton X-100 (PBS-T) for 15 mins at room temperature with rotation. After fixation,

ovaries were washed in PBS-T for 10 min 3 times. Next, ovarioles were carefully separated, mounted in Vectashield Antifade Mounting Medium (Vector Laboratories, Inc.; H-1000) and imaged on a Zeiss LSM880 confocal microscope (Carl Zeiss MicroImaging) with Nikon Plan-Apochromat 63x/1.40 NA oil immersion objective.

**Immunostaining.** For Fmr1 immunostaining, one-day-old GFP-Clu female flies were crossed with males in a ratio of 3:1 and fed yeast paste for two days. Flies were then starved overnight with tissue paper wet in water (starved flies) and refed with yeast paste for 2 h (refed flies). For ATP5A immunostaining, one-day-old flies were fed in yeast paste for 2 days, fasted overnight and refed for 6 h. Ovaries were dissected in Grace's media (Thermo Fisher Scientific; 11605045), fixed in 4% PFA made in 0.3% Triton X-100 PBS (PBS-T) for 15 min at room temperature with rotation and washed three times, 10 min with PBS-T. Subsequently, they were blocked in 1% BSA + PBS-T for 1 h, incubated with an antibody against Fmr1 (DSHB Hybridoma; 5A11) in 0.3% PBS-T containing 1% BSA, diluted 1:50 or an antibody against ATP5A (Sigma; ab14748), used at 1:500 dilution, overnight at 4 °C. Following the incubation with primary antibody, ovaries were washed three times, 10 min with PBS-T and incubated with the goat anti-mouse secondary antibody, Alexa Fluor 594 (Invitrogen; A11005) at 1:250 dilution made in 1% BSA PBS-T in the dark for 1 h. Next, ovaries were washes 3 times in PBS-T for 10 min and washed once in PBS. Ovarioles were separated, mounted in ProLong Diamond Antifade Mountant (Thermo Fisher Scientific; P36961) and imaged 24 h later in a Zeiss LSM 880 confocal microscope (Carl Zeiss MicroImaging) with a Nikon EC Plan-Neofluar 40x/1.30 Oil DIC M27 objective.

**Correlative light and electron microscopy (CLEM).** *Drosophila* ovaries expressing GFP-Clu were fixed with 4% formaldehyde (EM grade, Electron Microscopy Sciences; 15710) and 2.5% glutaraldehyde (EM grade; Sigma-Aldrich; G5882) in PBS for one h at room temperature, and then overnight at 4 °C. Next, they were washed three times in PBS. Ovarioles were transferred to a 35 mm dish (MatTek; P35GC-1.5-14-C) and immobilised with a small drop of low melting point agarose with coverslip (No. 1.5), left to set and then submerged in PBS. Confocal images were captured using an Olympus FV1000 confocal laser scanning microscope equipped with an UPlanSApo 60x/1.35 NA oil objective and zoom x3.5. Optical sections with a thickness of 250 nm were taken and the location and depth from the coverslip to positive labelled structures recorded. Deconvolution processing was performed using Huygens Essential version 21.04 (Scientific Volume Imaging, The Netherlands) with "Standard" profile settings. Following confocal imaging, samples were washed in 0.1 M cacodylate buffer for 1 h, and post-fixed in 1% osmium tetroxide/1.5% potassium ferricyanide in 0.1 M cacodylate buffer for 1 h. After further cacodylate washes, samples were stained with 1% tannic acid in 0.05 M cacodylate, followed by 5 min 'stop' in 1% sodium sulphate in 0.05 M cacodylate buffer. Samples were washed in water, and serially dehydrated through a 70, 90 & 100% ethanol series then propylene oxide, and finally flat embedded in TAAB 812 resin. Resin was polymerised at 60 ºC for 48 h. 400 nm thick sections were taken using a Reichert Ultracut E ultramicrotome to a depth of 4000 nm, and then 70 nm thick serial sections were cut and collected onto formvar coated slot grids. Sections were further contrasted with Reynold's lead citrate for 5 min before imaging at 120 kV on a JEOL JEM-1400 transmission electron microscope (TEM) (JEOL UK) using a Xarosa digital camera with Radius software (EMSIS, Germany). Low magnification TEM images (x1500) were roughly correlated to confocal images using characteristic features such as nuclei and cell walls to identify potential regions of interest. Each sequential section was imaged at the ROI at higher magnifications (2.5K and 5K) to perform CLEM. Optical sections were correlated to TEM images utilising characteristic features as fiducial markers using Fiji software [65] with TrakEM2 plugin [66]. All EM consumables sourced from Agar scientific ltd, UK, unless indicated otherwise.

**RNase treatment.** The RNase treatment was adapted from [36]. Briefly, ovaries from two-day-old GFP-Clu flies, kept with males and fed with yeast paste (n = 5) were dissected in Schneider's media (Lonza; LZ04-351Q) containing 200 µg/mL insulin (Sigma, I5500) (SI), then permeabilised in a buffer (100 mM potassium phosphate pH 7.5 and 0.2% Triton X-100) for 50 sec. Then ovaries were washed twice with SI and incubated either with SI only (control) or SI containing 300 µg/mL of RNase A (Ambion; AM2270) for 15 min. After washing ovaries twice with SI, samples were mounted in SI and immediately imaged in a Zeiss LSM 880 confocal microscope (Carl Zeiss MicroImaging) with a Nikon Plan-Apochromat 63x/1.40 NA oil immersion objective.

**Fluorescence oligo-(dT) in situ hybridisation.** Oligo-(dT) FISH was performed using a protocol described for third-instar larval salivary glands [67] with a few initial modifications. Egg chambers of two-day-old refed flies (4 h) were fixed as described in the *"Immunofluorescence"* section, washed 3 times in PBS-T and incubated 15 min with 1 U/µL Ribolock RNase Inhibitor (Thermo Fisher Scientific, EO0381) in 1XPBS on ice or with 300 µg/mL of RNase A (Ambion; AM2270) at room temperature. RNase inhibitor was added after the RNase treatment. Poly(A) RNA was stained with 5 ng/µL of an oligo-(dT)45-Alexa647 probe (Invitrogen).

**Cycloheximide and puromycin treatment.** One-day-old GFP-Clu female flies were crossed with males for one day in propionic food. Ovaries were dissected in Schneider's media and were either subjected to no treatment or treated with 100 µg/mL CHX (Sigma-Aldrich; C7698) or 50 µg/ml puromycin (MP Biomedicals; 11420802) for 10 mins. Then, 200 µg/mL insulin (Sigma-Aldrich; I5500) was added and incubated for 30 min. As an additional control, ovaries were maintained without any drug or insulin for 40 min. Ovaries were then fixed, mounted and imaged as described above (n > 3 flies for each condition).

**1,6-hexanediol treatment.** One-day-old GFP-Clu female flies were kept with males with yeast paste for two days. Ovaries from 5 flies were dissected in SI and placed onto a 35 mm dish (MatTek, P35GC-1.5-14-C) with SI. 4% of 1,6-hexanediol (Sigma-Aldrich; 240117) was added and snapshots were taken every min up to 15 min in a Zeiss LSM 880 confocal microscope (Carl Zeiss MicroImaging) with a Nikon Plan-Apochromat 63x/1.40 NA oil immersion objective.

## Yeast methods

**Strains and plasmids.** Yeast strains, plasmids and primers used in this study are detailed in S3 Table. The *CLU1* knockout strain was generated in the W303a parental strain through PCR-based gene disruption using a HYGMX4 cassette amplified from pAG32, as previously described [68]. Clu1-mCherry and Clu1-TEV-GFP were generated by integrating mCherry or TEV-GFP in the C-terminal of *CLU1* gene in the BY4741 strain using a PCR-based gene insertion as previously described [69]. BirA* was integrated into the BY4741 Clu1-GFP strain immediately following the GFP gene, and into the flanking regions of the His3 gene in BY4741, using the same PCR-based insertion approach. The mCherry cassette was amplified from pBS35 (Addgene plasmid; 83797), TEV-GFP from pFA6a-GFP(S65T) (Addgene plasmid; 41598) using a 5' primer containing the TEV sequence, and the BirA* cassette from pYM28-BirA* (Addgene plasmid; 160290). pRS313-pTDH3-Su9-TagBFP (mito-BFP) was generated using a yeast recombination-based cloning approach, replacing the *URA3* marker from pRS316-pTDH3-Su9-TagBFP (Addgene plasmid; 62383) by *HIS3*. *HIS3* and the flanking regions which are homologous to the pRS316 were amplified from a pRS413 plasmid and transformed into yeast with the NdeI linearised pRS316-pTDH3-Su9-TagBFP. Transformants were selected on SC without His plates and counter selected on SC without URA. Next, the plasmid was extracted from yeast using the QIAprep Spin Miniprep Kit (Qiagen; 27104) following the manufacturer instructions with an additional lysing step: after the addition of buffer P1, cells were lysed using acid-washed glass beads (425–600 µm diameter, Sigma-Aldrich; G8772) in a Precellys 24 homogenizer (Bertin Technologies), at 6,500 rpm for 15 sec with 5 min on ice, repeated 5 times. The extracted DNA was then transformed into DH5α competent cells (Invitrogen; 18265017) and transformed colonies were selected on LB containing Amplicillin. The correct deletion of *CLU1*, the insertion of mCherry, TEV-GFP, BirA* and the replacement of URA3 by HIS3 in pRS316-pTDH3-Su9-TagBFP were verified by colony PCR using primers that amplify across the flanking gene regions and by sequencing the inserts. BiFC strains were obtained from Bioneer (Korea). *CLU1-VN* strain (Mata) was mated with all the other VN strains (Matalpha) – *SOD2*, *PYC2*, *RPL17B*, *TIF3*, *ACO1*, *SCP160*, and *TIF5* to generate diploids, which were then selected in SC without uracil and leucine (SC-Ura-Leu). pRP2132, Ded1-mCherry [70] and pAG415GPD/Dcp1-dsRed were previously described [71].

**Media and growth conditions.** *Saccharomyces cerevisiae* yeast strains were grown on standard media, YPD or Synthetic Complete (SC) with all amino acids or without specific amino acids according to the plasmid selection. YPD contains 10 g/L yeast extract, 20 g/L peptone and 20 g/L dextrose. SC media includes 6.7 g/L of yeast nitrogen base with

ammonium sulphate and without amino acids, supplemented with a complete amino acid mixture or, a drop-out mixture according to the plasmid selection (concentrations of amino acid mixtures were based on manufacturer's instructions) and a carbon source. Depending on the auxotrophic markers, media was further supplemented with increased concentrations of histidine, leucine, methionine, uracil or adenine, to prevent amino acid starvation in the late log phase, as previously described [72]. Unless indicated otherwise, the carbon source used in most experiments was glucose (2%). Other carbon sources used were ethanol (3%), glycerol (3%) or galactose (3%). All media reagents were sourced from Formedium.

Unless otherwise specified, all cultures were initially grown as pre-cultures in glucose-containing SC media. These pre-cultures were then diluted back into the same media to synchronize cells in the log phase overnight. For media shifts, early log-phase cells were washed in SC media without any carbon source and resuspended in media containing a different carbon source or no carbon source for starvation experiments. All cultures were incubated in a shaking incubator set at 30 °C, 225 rpm.

The diauxic shift was determined by measuring glucose levels in the culture media using the Glucose (GO) Assay Kit (Sigma-Aldrich; GAGO20). The start of the PD phase was defined as the time when cells began regrowing, and oxygen consumption increased. Oxygen consumption was measured using an Oroboros oxygraphy-2K high-resolution respirometer (OROBOROS Instruments). Oxygen consumption rate was calculated by subtracting non-mitochondrial respiration (calculated after addition of 2 μM antimycin A) from the total oxygen consumption rate. Data was acquired and analysed using DatLab software (OROBOROS Instruments).

**Microscopy.** To immobilise live cells, 1% agarose solution (UltrapureTM Agarose, Invitrogen) in appropriate SC media was prepared. Agarose pads were created by pipetting 30 μl of heated agarose solution onto a slide in between two layers of sticky-tape. Then, a second slide was placed on top of the other and allowed to set. Right before imaging, the top slide was slowly slid off, resulting in an intact agarose pad attached to the bottom slide. Subsequently, 7 μl of cells were pipetted onto the pad and covered with a coverslip. Alternatively, cells were directly pipetted into a slide and imaged immediately. For yeast fixation, cultures were incubated in 4% formaldehyde (Thermo Fisher Scientific; 28908) solution containing 4% (w/v) sucrose made in PBS for 15 min at 30°C. After fixation, cells were washed twice in 1.2 M sorbitol and 0.1 M potassium phosphate buffer. Cells were mounted and immobilised in ProLong Diamond Antifade Mountant (Thermo Fisher Scientific; P36961) and imaged 24 h later.

Images were acquired with a Zeiss LSM 880 confocal microscope (Carl Zeiss MicroImaging) with Nikon Plan-Apochromat 63x/1.40 NA oil immersion objective, with a step size of 0.35 μm for z-stacks. Images were analysed with Fiji [65].

**Fluorescence recovery after photobleaching (FRAP).** Clu1-GFP cells were grown until PD phase and imaged using a Zeiss LSM880 Airyscan confocal microscope with a Plan-Apochromat 100x/1.4 NA oil objective. The selected regions of interest (ROI) were photobleached with a 488 nm laser at 100% laser intensity. Images were acquired from a single plane continuously (~3 s/frame) for 200 frames (~10 min). Fluorescence intensity over time was quantified in Fiji for three ROIs: the photobleached granule (I[bleach]), an unbleached granule in a neighbouring cell used as a reference control (I[nonbleach]), and a background region (I[background]). To account for background fluorescence and general photobleaching during acquisition, fluorescence intensities were normalized as previously described [73]:

$$\frac{(I[nonbleach]prebleach - I[background])}{(I[nonbleach]n - I[background])} x(I[bleach]n - I[background])$$

The corrected intensities were normalised to the pre-bleach intensities, setting the average of the pre-bleach values ($I[bleach]_{prebleach}$) to 1, and the first post-bleach value ($I[bleach]_0$) to 0 using the following equation:

$$\frac{(I[bleach]n - I[bleach]0)}{(I[bleach]prebleach - I[bleach]0)}$$

Data analysis and plotting was done in Microsoft Excel and GraphPad Prism 10.

**Stress treatments.** Mid-log cells grown in SC with 2% of glucose were subjected to the following treatments: heat shocked at 46 ºC for 30 min in a shaking heat block or maintained at 30 ºC as control; incubated with 1M potassium chloride for 30 min; with 3 mM hydrogen peroxide for 15 min; washed once with water and incubated in water for 10 min, or SC without any carbon source for 10 min; incubated with sodium azide 0.5% (v/v) for 15 min; treated with vehicle or 30 µM CCCP (prepared in 100% ethanol) for 15 min (unless otherwise stated) by direct addition into the media.

For stress tests in SC galactose, cells were grown as described above. After 6 h of incubation in galactose, when cells restarted growing, they were subjected to the same stress treatments as described for glucose-grown cells. Additionally, a mixture of 10 µM oligomycin and 40 µM antimycin A (OA) was tested for 30 min. For CHX tests, 100 µg/ml CHX (Sigma-Aldrich; C7698) (dissolved in 100% ethanol) or ethanol, as control, was added to Clu1-GFP cells in early PD cells (8 h after reaching optical density 5 at 600 nm absorbance ($OD_{600}$) in SC media with 2% glucose). Cells were imaged before and after adding CHX or ethanol. The same concentration of CHX or ethanol was also added to cells that were growing in SC galactose (3%) for 6 h (shifted from SC glucose media), followed by treatment with 30 µM CCCP for 15 min. Cells were fixed as described in the "*Microscopy*" section.

**1,6-hexanediol treatment.** Cells in the PD phase were treated with 10 µg/ml digitonin for 2 min with or without 5 or 10% 1,6-hexanediol for 5 min. They were immediately fixed and imaged as described in the "*Microscopy*" section.

**Proximity-dependent biotinylation identification (BioID).** Pre-cultures of BirA* and Clu1-GFP-BirA* strains were established in quadruplicate in SC-His media to synchronise growth. For mid-log phase samples, log phase cultures were diluted to reach an $OD_{600}$ of 2.5 after 16 h of incubation with 50 µM biotin. For PD phase samples, 50 µM biotin was added 8 h after reaching $OD_{600}$ 5, and incubated for 16 h. After biotin incubation, cells were centrifuged (4,000 rpm, 5 min), washed in sterile water and flash frozen in liquid nitrogen. Cells were lysed as previously described [74]. Briefly, 10% trichloroacetic acid was added to frozen cells (60 $OD_{600}$), incubated on ice for 20 min and subsequently centrifuged 3 min at 15,000 g, at 4 °C. The pellet was washed with ice-cold acetone twice, dried at room temperature and then resuspended with 750 µl of MURB buffer (50 mM sodium phosphate, 25 mM MES, pH 7.0, 1% SDS, 3 M urea, 0.5% 2-mercaptoethanol, 1 mM sodium azide). Next, the pellet was disrupted by vortex with acid-washed glass beads (425–600 µm diameter, Sigma-Aldrich; G8772) in Precellys 24 homogeniser (Bertin Technologies), at 6,500 rpm for 15 sec with 5 min on ice, repeated 5 times. Next, the lysate was centrifuged at 18,000 g at 4 °C for 15 min and the supernatant transferred into a new tube.

For affinity purification, 600 µl Pierce high-capacity streptavidin agarose Resins beads (Thermo Fisher Scientific; 20359) were first equilibrated by washing three times with affinity purification (AP) buffer (8 M Urea, 1% SDS, 50 mM Tris-HCL Buffer (pH 7.4), 1 mM DTT, cOmplete EDTA-free Protease Inhibitor Cocktail (Roche)). After equilibration, the cell lysate was diluted to 7.5 ml with the AP buffer. Lysates were incubated with the equilibrated beads overnight, at room temperature. The following day, the beads were washed 5 times by centrifugation at 1,000 g for 2 min and resuspending beads in wash buffer (8 M Urea, 1% SDS, 50 mM Tris-HCL Buffer (pH 7.4), 1 mM DTT). Biotinylated proteins were eluted in 750 µl Laemmli Sample Buffer (Bio-Rad; 1610747) containing β-mercaptoethanol (Sigma-Aldrich; M6250) incubating it at 95 °C for 10 min. Beads were removed from eluates using Vivaclear 0.8 µm PES centrifugal microfilters (Sartorius; VK01P042) and eluates were concentrated via Vivaspin 500 centrifugal concentrators (Merck; Z614068) with molecular weight cut-off of 30 kDa to remove free streptavidin. Proteins were resolved by SDS-PAGE and 6 gel regions from each sample were excised. Regions of gel containing a prominent 250 kDa protein, present in every sample, were excluded from the analyses.

Proteins were trypsinised within the gel. The tryptic peptide mixtures were desalted and purified by C18-reverse phase chromatography (ZipTip, Millipore) and then analysed by LC-MS/MS using a Proxeon EASY-nanoLC system, with a C18 column (50 µm x 150 mm) coupled directly to a Q-Exactive plus Orbitrap mass spectrometer (Thermo Fisher Scientific). Peptides were fragmented by collision induced dissociation with nitrogen. Raw MS data were analysed using MaxQuant software [75] as described in [76]. Data was searched against a sequence database of

*Saccharomyces cerevisiae* proteins (UniProt) that was modified to include streptavidin sequence. Relative protein abundances were compared by label-free quantification. Data normalisation and statistical analysis (t-test) were performed in Perseus software [77].

**Puromycin-associated nascent chain proteomics (Punch-P).** Punch-P was performed as described in [47,48,76]. Wild-type W303 (WT) and *clu1Δ* cells in the log phase were shifted from SC + glucose to SC + ethanol 3% and grown for 4 h (n = 4). Yeast cells equivalent to 300 OD$_{600}$ per condition were pelleted, washed once with cold water, snap-frozen, and stored at −70 °C. Each cell pellet was thawed and resuspended in 900 µl of polysome buffer (PLB) (20 mM Tris-HCl, pH 7.4, 140 mM KCl, 10 mM MgCl$_2$, 0.5 mM DTT, 40 U/mL RNasin (Promega, N2111), 1.4 µg/mL pepstatin (Sigma-Aldrich; 10253286001), 2 µg/mL leupeptin (Sigma-Aldrich; 11017101001), 0.2 mg/mL heparin (Sigma-Aldrich; H3393), EDTA-free cOmplete protease inhibitor mix (Roche; 4693159001)) containing 1% Triton X-100. Cells were lysed using acid-washed glass beads (425–600 µm diameter, Sigma-Aldrich; G8772) corresponding to an approximate volume of 300 µL and homogenised in Precellys 24 (Bertin Technologies) at 6,500 rpm for 10 sec with 5 min on ice in between, repeated 5 times. Cell debris was removed by centrifuging the lysates at 17,400 g for 30 min at 4 °C. Next, 670 µL of lysate was slowly dispensed on top of 330 µL of a 70% sucrose cushion in a 1 mL polycarbonate tube (Beckman Coulter; 343778) and centrifuged in an MLA-130 ultracentrifuge rotor (Beckman Coulter) at 48,600 g for 160 min at 4 °C. The ribosome pellets were gently washed in 500 µl of ice-cold RNase-free water to remove all sucrose. Polysomes were carefully resuspended in 90 µl PLB (without detergent) until a homogeneous, slightly opaque solution was obtained. rRNA absorbance was measured at 254 nm, and the volume corresponding to 15 OD$_{600}$ was incubated with 1.5 nmol of Biotin-dC-puromycin (Jena Bioscience; NU-925-BIO-S) at 37 ºC for 15 min to label nascent peptides. The same volume of polysomes was kept aside and used as control to identify endogenously biotinylated proteins and other contaminants (unlabelled). Labelled and unlabelled samples were incubated with 75 µl of Pierce Streptavidin Agarose (Thermo Fisher Scientific; 20347) in 1 mL of SDS/urea buffer (50 mM Tris-HCl, pH 7.5, 8 M urea, 2% SDS, and 200 mM NaCl) overnight at room temperature. Next day, beads were washed five times with SDS/urea buffer and incubated in 1 M NaCl for 30 min, followed by five washes in ultrapure water, 30 min incubation with 1 mM DTT, 30 min incubation with 50 mM iodoacetamide and two washes with 50 mM ammonium bicarbonate. Washes were achieved by centrifugation at 1,000 g for 2 min. Proteins were trypsinised on beads and identified by LC-MS/MS analysis as described in the "*BioID*" section. Proteins more abundant in both WT and *clu1Δ* unlabelled samples were considered contaminants.

**Calculation of translation efficiency.** Translation efficiency was quantified by dividing the levels of newly synthetised proteins found by Punch-P (label-free quantification) by the steady-state levels of the corresponding mRNA, which were determined by RNA-seq data.

**Clu1 RNA immunoprecipitation (RIP).** Clu1-GFP and cells expressing GFP via p413-GPD-GFP were grown until the mid-log phase; cells equivalent to 50 OD$_{600}$ were pelleted, flash-frozen and stored at -70 ºC. Cells were resuspended in 800 µL of PLB (20 mM HEPES pH 7, 150 mM KCl, 5 mM MgCl$_2$, 0.5% NP-40, 1 mM DTT, 1 mM PMSF, 400 U/mL RNaseOUT (Thermo Fisher Scientific; 10777019) 100 µg/mL CHX (Sigma-Aldrich; C7698-1G), EDTA-free cOmplete protease inhibitor mix (Roche; 4693159001)) and lysed in Precellys 24 (Bertin Technologies) as described in the "*Punch-P*" section. For puromycin and EDTA-treated samples, CHX was omitted from PLB. For EDTA-treated samples, EDTA was added to lysates at 50 mM concentration. After lysis, cells were centrifuged at 3,000 rpm twice for 10 min at 4 ºC. Lysates were pre-cleared with Dynabeads protein G beads (Thermo Scientific; 10003D) for 30 min, and then 25 µl of the lysate was taken for RNA extraction (input). Precleared lysates were incubated with the Dynabeads conjugated with a mouse monoclonal GFP antibody (Abcam; ab1218) for 2 h 30 min at 4 ºC, using an end over end rotator. Next, beads were washed 6 times with wash buffer (20 mM HEPES pH 7, 350 mM KCl, 5 mM MgCl$_2$, 1%, NP-40, 0.1 mM DTT, 100 µg/mL CHX). CHX was not added to the wash buffer of samples treated with EDTA or puromycin. For puromycin treated samples, after the last wash, beads

were resuspended with PLB containing 500 mM of KCl instead of 350 mM and 1 mM puromycin and incubated for 10 min at 37 ºC. All samples were then resuspended with wash buffer containing 10 U of DNase I (Thermo Scientific; AM2235), incubated for 2 min at 37 °C, and washed one more time with wash buffer. Finally, beads were resuspended in Tri reagent LS (Sigma-Aldrich; AM9738) to extract mRNAs. All buffer reagents were purchased as Molecular Biology, Nuclease-Free grade.

**Reverse transcription-quantitative real-time PCR (RT-qPCR).** RNA was extracted using Direct-zol RNA microprep kit (Zymo Research; R2060) following the manufacturer's instructions. RNA extracted from polysome pellets (Punch-P) was treated with TURBO DNA-free Kit (Thermo Fisher Scientific; AM1907) to remove DNA contamination following the manufacturer's instructions. cDNA was synthetised using Maxima H Minus cDNA Synthesis Kit with dsDNase (Thermo Fisher Scientific; M1681) following the manufacturer's instructions. For polysome pellet samples, 500 ng of total RNA was used for cDNA synthesis, while for RNA extracted from RIP or input samples, the whole samples were used. cDNA was amplified using PowerUp SYBR Green Master Mix for qPCR (Thermo Fisher Scientific; A25741) following manufacturer's instructions, in a QuantStudio 3 Real-Time PCR System (Applied Biosystems). Primers used are described in S3 Table. Relative quantification was performed using the comparative CT method using PCR primer efficiency [78]. For Clu1 RIP, mRNA levels of *ACO1*, *PDA1*, *HSP60, SSC1*, *ATP1*, *ADH3*, *ATP2*, *SOD2, TAF10 and UBC6* were quantified by RT-qPCR from RIP and input lysate samples. To normalise for small variations in cell lysis, RIP mRNA levels were divided by the corresponding input levels for each sample. To correct for bead loss during washes or other technical variability, mRNA levels of *ACO1*, *PDA1*, *HSP60, SSC1*, *ATP1*, *ADH3*, *ATP2*, *SOD2* and 18S and 25S rRNA detected in RIP were normalised to the geometric mean of non-interactor controls, *TAF10 and UBC6*. Finally, Clu1-GFP RIP mRNA levels were normalised by control (GFP) RIP levels.

**Clu1 immunoprecipitation.** Cells expressing Clu1-TEV-GFP or GFP (p413-GPD-GFP) were harvested in the log phase and immunopurified as described in the "*Clu1 RIP*" section (PLB with CHX). After the last wash, an aliquot of beads to be analysed by WB (IP) was stored and the remaining beads were incubated overnight at 4 ºC with 100 µl PLB containing 0.5 µg of EDTA-free TEV protease (Sigma; T4455) to release Clu1 and co-interactors from the beads (eluate). Finally, the eluates were concentrated in Vivaspin 500 centrifugal concentrators with 10 kDa cut-off (Merck; Z614033). To elute proteins from IP samples, beads were incubated with 2x Laemmli Sample Buffer (Bio-Rad; 1610747) containing 1:10 β-mercaptoethanol (M6250; Sigma-Aldrich) for 10 min at 95 ºC.

**Sucrose gradient ultracentrifugation.** Clu1-GFP yeast cells were analysed in different growth conditions: 1) mid-log phase, 2) PD, 3) grown until early log in SC glucose, shifted to SC galactose and treated with CCCP as described in the "*Stress treatments*" section and 4) grown as in condition 3) and spiked with 2% glucose and incubated for 15 min. Cells were cross-linked with formaldehyde in all conditions, with exception of mid-log cells, which were also analysed without cross-linking. For cross-linking, media was removed by centrifugation at 4,000 rpm for 5 min, resuspended in 1% formaldehyde made in PBS and incubated with rotation for 20 min at room temperature. To quench the reaction, 125 mM of glycine was added for 5 min. Cells were then washed once with PBS, pelleted, snap-frozen with liquid nitrogen and stored at -70 ºC. Cells equivalent to 50 $OD_{600}$ were used for all conditions.

Frozen cell pellets were lysed in 600 µl of the same polysome buffer (PLB) described in the "*Clu1 RIP*" section, with the following modifications: 15 U of TURBO DNase (Thermo Fisher Scientific; AM2238) were added to PLB; for EDTA non-cross-linked treated samples, CHX was excluded from the buffer; for RNase treated samples, RNaseOUT was excluded from the buffer. Lysis was performed as described in the "*Punch-P*" section and debris was removed by spinning lysate twice at 3,000 rpm at 4 ºC for 10 min. For EDTA non-cross-linked samples, 50 mM EDTA was added after lysis and for RNase-treated samples, 1 µg of RNase A (Ambion; AM2270) was added per 600 µg of RNA and incubated for 30 min at room temperature. To stop the RNase A reaction, 200 U of RNaseOUT (Thermo Fisher Scientific; 10777019) were added for each µg of RNase A used. Sucrose continuous density gradients, 5–45% sucrose in PLB buffer (without RNaseOUT or DNase), were prepared using a Gradient Station (BioCOMP). For EDTA samples, 50 mM EDTA was also added to the

buffers and CHX was excluded. Clarified lysates were layered on top of the gradients, ultracentrifuged in a SW 40 Ti rotor (Beckman Coulter) at 39,300 rpm, 4 °C, for 2 h 30 min, and fractionated using a Gradient Fractionator (Brandel). RNA profiling across gradients was performed by continuous monitoring of absorbance at 254 nm using an AKTA prime plus (Cytiva), which allowed the identification of fractions enriched in ribosomal subunits, monosomes and polysomes (labelled in the immunoblots).

**Immunoblotting.** Proteins were separated by SDS–PAGE in 4–20% Mini-PROTEAN TGX Precast Protein Gels (Bio-Rad; 4561096) and then transferred onto a nitrocellulose membrane (Bio-Rad; 1704158) using the Trans-Blot Turbo Transfer System. Membranes were blocked for 1 h at room temperature with 5% (wt/vol) dried skimmed milk powder (Marvel Instant Milk) in TBS containing 0.1% Tween-20 (TBS-T). Following blocking, membranes were incubated overnight at 4°C with the appropriate primary antibodies diluted in TBS-T with exception of anti-Pgk1, which was only incubated for 1 h. After three washes with TBS-T for 10 min, membranes were incubated for 1 h at room temperature with HRP-conjugated secondary antibodies diluted in 5% milk in TBS-T. Membranes were then washed four times, for 10 min in TBS-T, and antibody detection was performed using the Amersham ECL Prime detection kit (Cytiva; RPN2232). Sucrose gradient immunoblots were stripped after incubation with anti-Rpl3 with Restore Western Blot Stripping Buffer (Thermo Fisher Scientific; 21059) according to the manufacturer's instructions before incubating with anti-Pgk1. The immunoblots were imaged with the Amersham Imager 680.

The primary antibodies used for immunoblotting in this study were: anti-GFP (1:1000; Abcam; ab290), anti-BirA (1:1000; Novus Biologicals; NBP2–59939), anti-Pgk1 (1:10000; Thermo Fisher Scientific; 459250) and anti-Rpl3 (1:2000; DHSB; ScRPL3). Secondary antibodies used were: HRP-conjugated goat anti-mouse IgG H&L (1:20000; Abcam; ab6789) and HRP-conjugated goat anti-rabbit IgG H&L (1:10000; Thermo Fisher Scientific; G-21234).

**Yeast spotting assay.** WT and *clu1Δ* strains were grown in SC containing glucose and diluted once back to synchronize cells to early log phase. Cell concentration was adjusted to be equal among strains ($OD_{600} = 1$) and 5-fold serial dilutions were prepared with water in a 96-well plate. Next, they were spotted onto SC containing 2% glucose or 3% ethanol plates. Plates were left in the 30 ºC incubator until growth was observed. Images were obtained with a GelDoc XRS+ (BioRad).

## Statistical analysis

GraphPad Prism 9 (SCR_002798; RRID) was used to perform all statistical analyses. Two-sample groups were analysed by unpaired t-test with Welch's correction. Multiple sample groups were compared using one-way ANOVA with Tukey's post-hoc test.

## Supporting information

**S1 Fig. Clu/Clu1 punctate subcellular distribution in several fly tissues and yeast growing in glucose containing media.** (A) Confocal imaging of GFP-Clu third instar larval and two-day-old adult tissues. VNC, ventral nerve cord. Scale bars: 40 μm (top panels) and 8 μm (bottom panels, magnified images of boxed areas). (B) Maximum intensity projection of an egg chamber of a three-day-old female fly refed for 6 h. The white box indicates the region used for 3D rendering in C. (D, E) Box plots representing the area (D) and volume (E) of Clu foci from the region used for 3D rendering. The plots display the minimum, first and third quartile, median and maximum values. (F) Graph showing Clu1-GFP cells' growth in glucose-containing media, percentage of cells containing Clu1-GFP foci and basal respiration throughout growth. Cells undergo a diauxic shift upon exhaustion of glucose from the media. During this time, cells cease growth and suffer a drastic transcriptomic and proteomic shift to adapt to the new carbon source available in the media, ethanol. They transition from a fermentative to a respiratory metabolism. We considered the start of PD phase as the moment cells started regrowing and respiration increased.
(TIF)

**S2 Fig. Clu1 protein levels remain stable after the formation of granules.** Immunoblot analysis of Clu1-GFP and Pgk1 in Clu1-GFP-expressing cells, either not treated (NT) or treated with sodium azide (NaN₃) to induce granule formation. The graph indicates the Clu1-GFP immunoblot quantification normalised by Pgk1 (mean ± SEM; n = 3; paired t-test; ns = non-significant).
(TIF)

**S3 Fig. *Drosophila* GFP-Clu does not colocalise with SG or PB markers.** (A, B) Confocal microscopy of GFP-Clu egg chambers fasted (16 h) or refed (6 h) and immunostained for (A) Fmr1 (SG marker) or (B) co-expressing Tral-mRFP (PB marker). Scale bars = 20 µm, inset = 4 µm.
(TIF)

**S4 Fig. Localisation of Clu1 and the effects of its absence on growth and mitochondrial morphology.** (A) Confocal image of Clu1-GFP cells expressing mito-mCherry in the PD phase. Inset box indicates zoomed image, with the intensity profile plot along the dashed line for Clu1-GFP and mito-mCherry fluorescence. (B, C) Spotting assay of *clu1Δ* and the parental strain W303A. Mid-log phase cells grown in glucose-containing media were adjusted to the same optical density, five-fold serial diluted and spotted onto media containing either glucose (B) or ethanol (C) as carbon sources. (D, E) *clu1Δ* and wild-type cells expressing mito-mCherry were imaged by confocal microscopy in media containing either glucose (D) or ethanol (E). Scale bars = 5 µm, inset = 1.25 µm.
(TIF)

**S5 Fig. Validation of Clu1-GFP-BirA\* strain.** (A) Immunoblot of BirA\* and Clu1-GFP-BirA\* strains grown in the log and PD phases, with Pgk1 as loading control. (B) Confocal images of Clu1-GFP and Clu1-GFP-BirA\* cells in the log and PD phases. Scale bar = 5 µm.
(TIF)

**S6 Fig. Loss of Clu1 increases sensitivity to acute heat shock during metabolic adaptation.** Spotting assay of *clu1Δ* and wild-type cells. Cultures were grown to early log phase in glucose-containing media at 30 °C, then heat shocked at 50 °C for 20 minutes or left at 30 °C (no heat shock), under three conditions: from original growth conditions (Glucose (log)), following a shift to ethanol-containing media during the lag phase while cells were adapting to respiration (Ethanol (lag)), or after cells had resumed growth (Ethanol (log)). Cultures were normalised to the same optical density, serial diluted, and spotted onto glucose-containing media to assess survival.
(TIF)

**S7 Fig. Translation efficiency of proteins with altered translation in the *clu1Δ*.** Graphs indicate the translation efficiency for mRNAs whose translation was reduced (A) or increased (B) in *clu1Δ* strain compared to wild type. Translation efficiency was calculated as the ratio of nascent protein levels (Punch-P) to the steady-state levels of their corresponding mRNA (RNAseq) and normalised to the wild-type levels.
(TIF)

**S1 Table. Label-free quantification and hit summary of BioID analysis.**
(XLSX)

**S2 Table. Label-free quantification and hit summary of Punch-P analysis.**
(XLSX)

**S3 Table. Strains, plasmids and primers used in the study.**
(XLSX)

**S4 Table. Raw data and summary statistics for all graphs.** The data for every graph in both the main text and supplementary material is listed within individual sheets. Sheets are labelled by the Figure number and panel.
(XLSX)

**S1 Video. Addition of glucose to Clu1-GFP cells in PD phase.** Clu1-GFP cells in the PD phase were layered on an agarose pad made with media containing 2% glucose and imaged over a 30 min time course.
(AVI)

**S2 Video. Addition of insulin to GFP-Clu egg chambers.** Time course over 62 min, imaged at one-min intervals, shows dissected egg chambers from two-day-old GFP-Clu flies following the addition of insulin.
(AVI)

## Acknowledgments

We kindly thank Rachel Cox (Uniformed Services University), Daniel St Johnston (University of Cambridge) and Erika Geisbrecht (Kansas State University) for generously sharing fly lines, and Nianshu Zhang (University of Cambridge), Roy Parker (University of Colorado Boulder), Benjamin Glick (University of Chicago), Jonathan Weissman (Whitehead Institute), Eric Muller (University of Washington), John Pringle (Stanford University) and Martin Ott (University of Gothenburg) for kindly sharing yeast strains and plasmids. We thank Ian Fearnley and Shujing Ding (MRC Mitochondrial Biology Unit) for technical assistance with the mass spectrometry, Roy Chowdhury (MRC Mitochondrial Biology Unit) for help setting up the FRAP method and Natalie Allcock, Anna Straatman-Iwanowska and Kees Straatman (University of Leicester) for the CLEM work. Finally, we thank Jeffrey Gerst (Weizmann Institute of Science) for technical assistance, Brian Zid and Yuko Sugiyama (University of California, San Diego) and all members of the Whitworth lab for discussions during the project and feedback on the manuscript.

## Author contributions

**Conceptualization:** Leonor Miller-Fleming, Alexander J. Whitworth.

**Formal analysis:** Leonor Miller-Fleming, Wing Hei Au, Pedro Rebelo-Guiomar, Jasper Schmitz.

**Funding acquisition:** Leonor Miller-Fleming, Alexander J. Whitworth.

**Investigation:** Leonor Miller-Fleming, Wing Hei Au, Laura Raik, Jasper Schmitz, Ha Yoon Cho, Aron Czako, Alexander J Whitworth.

**Methodology:** Leonor Miller-Fleming, Wing Hei Au, Pedro Rebelo-Guiomar, Jasper Schmitz, Ha Yoon Cho.

**Project administration:** Alexander J. Whitworth.

**Resources:** Alexander J Whitworth.

**Supervision:** Leonor Miller-Fleming, Pedro Rebelo-Guiomar, Alexander J. Whitworth.

**Validation:** Leonor Miller-Fleming, Wing Hei Au, Laura Raik, Pedro Rebelo-Guiomar.

**Visualization:** Leonor Miller-Fleming, Pedro Rebelo-Guiomar.

**Writing – original draft:** Leonor Miller-Fleming, Alexander J. Whitworth.

**Writing – review & editing:** Leonor Miller-Fleming, Wing Hei Au, Pedro Rebelo-Guiomar, Alexander J. Whitworth.

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
