## [Decision Letter · Decision Letter 0]

Dear Dr Whitworth,

We are pleased to inform you that your manuscript entitled "Clu1/Clu form mitochondria-associated granules upon metabolic transitions and regulate mitochondrial protein translation via ribosome interactions" has been editorially accepted for publication in PLOS Genetics. Congratulations!

Yours sincerely,

Joseph Opferman, PhD

Academic Editor

PLOS Genetics

Pablo Wappner

Section Editor

PLOS Genetics

Aimée Dudley

Editor-in-Chief

PLOS Genetics

Anne Goriely

Editor-in-Chief

PLOS Genetics

Comments from the reviewers (if applicable):

Reviewer's Responses to Questions

Comments to the Authors:

Please note here if the review is uploaded as an attachment.

Reviewer #1: The authors did a reasonable job in responding to the reviewer comments. Some questions could not be answered for personnel or technical reasons and the authors chose to remove the data in question. Given that the localization of Clu to mitochondrial outer membrane is a crucial data supporting the regulatory role of Clu in the translation of nuclear encoded mitochondrial mRNAs, the data in Figure 3 should be quantified using colocalization coefficient, etc. Moreover, the treatment of cells with 1,6-hexanediol in Figure 4 lacks critical controls: does 1,6-hexanediol have any non-specific effect in disassembling protein assemblies or degrading Clu-GFP? Looking at 1,6-hexanediol effect on other protein assembly not depending on phase separation, and looking at Clu-GFP protein level before and after 1,6-hexanediol treatment should be presented.

Reviewer #2: Summary and Overall Evaluation

This manuscript presents a comprehensive and well-executed study of Clu/Clu1 granules in Drosophila melanogaster and Saccharomyces cerevisiae, with comparative insights into their mammalian homolog, CLUH. The authors characterize the granules’ molecular dynamics and provide strong evidence for their model that Clu/Clu1 granules sequester ribosomes engaged in translating specific mRNAs in response to metabolic changes. The study is logically structured, well written, and contributes significantly to our understanding of RNA granules and mitochondrial regulation.

Evidence Reproducibility and Clarity

The authors convincingly demonstrate that Clu/Clu1 granules localize dynamically in response to the metabolic state of the cell. Through a series of complementary experiments, they distinguish these granules from other biomolecular condensates by showing that Clu1 foci form specifically under a subset of mitochondrial stress conditions and do not colocalize with other known condensates. Using correlative light and electron microscopy, they define the granules as membraneless and adjacent to mitochondria and confirm this structure through sensitivity to 1,6-hexanediol.

The role of RNA is well supported: RNA is enriched in the granules, protected from RNase in fixed cells, and necessary for granule maintenance in live cells. The dependence on active translation is demonstrated by differential responses to puromycin and cycloheximide. Protein-protein interactions identified through BioID and validated by fluorescence complementation suggest Clu1 interacts with ribosomal and mitochondrial proteins. Finally, comparisons between wild-type and clu1∆ cells support a role for granules in coordinating mitochondrial protein synthesis, and ribosome association is confirmed to be independent of other cofactors.

Overall, the data presented is robust, the logic is clear, and the manuscript succeeds in connecting these findings to broader themes in RNA biology and mitochondrial regulation.

Comments

1.) Evolutionary Context of Protein Homology:

While the study emphasizes functional conservation across divergent eukaryotes, it lacks a detailed analysis of sequence- or structure-level conservation among Clu, Clu1, and CLUH. A comparison of primary sequences, domain architectures, or structural models would enhance the evolutionary narrative and substantiate the functional parallels being drawn.

2.)Role of Mitochondrial Morphology:

In Figures 2 and 3, mitochondrial stress (e.g., via sodium azide treatment) appears to correlate with changes in mitochondrial organization. It is unclear whether these changes contribute to, or result from, Clu/Clu1 granule formation. Clarifying this relationship—or explicitly noting its uncertainty—would improve the interpretation of these experiments.

Significance

This study offers a detailed characterization of Clu/Clu1 granules and their role in translation regulation in response to metabolic cues. By demonstrating conserved functional features in both yeast and fly, and referencing mammalian parallels, the authors broaden the relevance of their findings. This work will be of interest to researchers studying mitochondrial function, metabolism, RNA granules, and the evolution of post-transcriptional regulation. The findings are novel, well supported, and thoughtfully contextualized.

Reviewer #3: The manuscript by Miller-Fleming et al. describes the role of Clu1 (S. cerevisiae) and Clu (D. melanogaster), homologues of mammalian CLUH protein, in the translation of several nuclear encoded mitochondrial proteins. Clu1/Clu form granules under different metabolic conditions, that are different from stress granules or P-bodies. Clu1/Clu bind several mRNA coding for mitochondrial proteins, while these are actively translated on the cytosolic ribosomes, regardless of whether Clu1/Clu is present in a diffuse or granular state. The data presented here show that the function of Clu1/Clu is conserved throughout evolution, as they confirm previously reported findings for mammalian CLUH.

The study was performed with all due diligence, including validation for both homologues when possible. It is clearly presented and well explained. The authors addressed previously raised concerns and added additional data to strengthen the results presented in the manuscript.

I only have one question/comment: As Clu1-BioID analysis identified several mitochondrial proteins as proximity interactors, is it possible that these proteins are stored in Clu1 granules prior of their import into mitochondria? The ability of Clu1 to biotinylate the newly synthetized proteins implies that they are not co-translationally imported into mitochondria, or is Clu1 possibly residing at the exit tunnel of the ribosome?

Minor comment:

The authors were very careful in describing “the translation of nuclear-encoded mitochondrial proteins” rather than simply stating “mitochondrial protein translation” as the latter generally refers to the translation of mitochondrial-encoded proteins. However, one sentence should be revised: “Altogether, these findings suggest that Clu1 plays a critical role in coordinating mitochondrial protein translation, which is essential for metabolic adaptation and stress response.” to “… coordinating protein translation of nuclear-encoded mitochondrial proteins….”

Have all data underlying the figures and results presented in the manuscript been provided?

Reviewer #1: Yes

Reviewer #2: Yes

Reviewer #3: Yes

PLOS authors have the option to publish the peer review history of their article (what does this mean? ). If published, this will include your full peer review and any attached files.

**Do you want your identity to be public for this peer review?** For information about this choice, including consent withdrawal, please see our Privacy Policy .

Reviewer #1: No

Reviewer #2: No

Reviewer #3: No

Data Deposition

http://datadryad.org/submit?journalID=pgenetics&manu=PGENETICS-D-25-00553

Press Queries

---

## [Editor Report · Acceptance letter]

PGENETICS-D-25-00553

Clu1/Clu form mitochondria-associated granules upon metabolic transitions and regulate mitochondrial protein translation via ribosome interactions

Dear Dr Whitworth,

We are pleased to inform you that your manuscript entitled "Clu1/Clu form mitochondria-associated granules upon metabolic transitions and regulate mitochondrial protein translation via ribosome interactions" has been formally accepted for publication in PLOS Genetics! Your manuscript is now with our production department and you will be notified of the publication date in due course.

With kind regards,

Livia Horvath

PLOS Genetics

On behalf of:
